

# Anaerobic oxidation of methane alters sediment records of sulfur, iron and phosphorus in the Black Sea

Matthias Egger[1], Peter Kraal[1], Tom Jilbert[1,2], Fatimah Sulu-Gambari[1], Célia J. Sapart[3,4], Thomas Röckmann[3] and Caroline P. Slomp[1]

[1]Department of Earth Sciences – Geochemistry, Faculty of Geosciences, Utrecht University, P.O. Box 80021, 3508 TA, Utrecht, The Netherlands
[2]Now at: Department of Environmental Sciences, Faculty of Biological and Environmental Sciences, University of Helsinki, P.O. Box 65 (Viikinkaari 2a), 00014 Helsinki, Finland
[3]Institute for Marine and Atmospheric Research Utrecht (IMAU), Utrecht University, Princetonplein 5, 3584 CC Utrecht, The Netherlands
[4]Laboratoire de Glaciologie, Université Libre de Bruxelles, Belgium

*Correspondence to*: Matthias Egger (m.j.egger@uu.nl)

**Abstract.** The surface sediments in the Black Sea are underlain by extensive deposits of iron (Fe) oxide-rich lake sediments that were deposited prior to the inflow of marine Mediterranean Sea waters ca. 9000 years ago. The subsequent downward diffusion of marine sulfate into the methane-bearing lake sediments has led to a multitude of diagenetic reactions in the sulfate-methane transition zone (SMTZ), including anaerobic oxidation of methane (AOM) with sulfate. While the sedimentary cycles of sulfur (S), methane and Fe in the SMTZ have been extensively studied, relatively little is known about the diagenetic alterations of the sediment record occurring below the SMTZ. Here we combine detailed geochemical analyses of the sediment and pore water with multicomponent diagenetic modeling to study the diagenetic alterations below the SMTZ at two sites in the western Black Sea. We focus on the dynamics of Fe, S and phosphorus (P) and demonstrate that diagenesis has strongly overprinted the sedimentary burial records of these elements. Our results show that sulfate-mediated AOM substantially enhances the downward diffusive flux of sulfide into the deep limnic deposits. During this downward sulfidization, Fe oxides, Fe carbonates and Fe phosphates (e.g. vivianite) are converted to sulfide phases, leading to an enrichment in solid phase S and the release of phosphate to the pore water. Below the sulfidization front, high concentrations of dissolved ferrous Fe ($Fe^{2+}$) lead to sequestration of downward diffusing phosphate as authigenic vivianite, resulting in a transient accumulation of total P directly below the sulfidization front.

Our model results further demonstrate that downward migrating sulfide becomes partly re-oxidized to sulfate due to reactions with oxidized Fe minerals, fueling a cryptic S cycle and thus stimulating slow rates of sulfate-driven AOM ($\sim 1 - 100$ pmol cm$^{-3}$ d$^{-1}$) in the sulfate-depleted limnic deposits. However, this process is unlikely to explain the observed release of dissolved $Fe^{2+}$ below the SMTZ. Instead, we suggest that besides organoclastic Fe oxide reduction, AOM coupled to the reduction of Fe oxides may also provide a possible mechanism for the high concentrations of $Fe^{2+}$ in the pore water at depth. Our results reveal that methane plays a key role in the diagenetic alterations of Fe, S and P records in Black Sea sediments. The downward sulfidization into the limnic deposits is enhanced through sulfate-driven AOM with sulfate and AOM with Fe oxides may provide a deep source of dissolved $Fe^{2+}$ that drives the sequestration of P in vivianite below the sulfidization front.



## 1 Introduction

Anaerobic oxidation of methane (AOM), a process initially regarded as a biogeochemical curiosity, functions as an
important sink for oceanic methane ($CH_4$) by consuming > 90 % of all $CH_4$ produced in marine sediments (Knittel
and Boetius, 2009; Reeburgh, 2007). Although recent studies indicate that the biological oxidation of $CH_4$ could be
coupled to various additional electron acceptors such as nitrate and nitrite (Ettwig et al., 2010; Raghoebarsing et al.,
2006) as well as metal oxides (Beal et al., 2009; Egger et al., 2015b; Riedinger et al., 2014; Scheller et al., 2016;
Segarra et al., 2013; Sivan et al., 2011), sulfate ($SO_4^{2-}$) is commonly thought to be the dominant electron acceptor in
anoxic marine systems (Knittel and Boetius, 2009; Reeburgh, 2007).
Nevertheless, a coupling between anaerobic $CH_4$ oxidation and iron (Fe) oxide reduction (Fe-AOM) could have a
significant impact on sedimentary Fe cycling and related processes such as phosphorus (P) diagenesis, because of the
8:1 Fe-$CH_4$ stoichiometry of the reaction (Beal et al., 2009; Egger et al., 2015a; Rooze et al., 2016). Environmental
conditions that favor Fe-AOM in marine systems are still poorly understood. The required co-occurrence of pore
water $CH_4$ and abundant reducible Fe oxides suggests that Fe-AOM may occur in sediments that receive a relatively
high input of Fe oxides compared to the in-situ production of sulfide, which could allow a portion of Fe oxides to
escape the conversion to authigenic Fe sulfides and to remain preserved in the methanogenic sediments below the
zone of $SO_4^{2-}$ reduction (Egger et al., 2015b; Riedinger et al., 2014; Rooze et al., 2016). In addition, perturbations
inducing transient diagenesis such as anthropogenic eutrophication or climate change may also create diagenetic
environments that are likely favorable for Fe-AOM, as they provide a mechanism for the burial of Fe oxide-rich
deposits below sulfidic sediment layers (Egger et al., 2015b; Riedinger et al., 2014).
The Black Sea represents a good example of a sedimentary system in which transient diagenesis associated with
postglacial sea-level rise has led to the accumulation of sulfidic sediments above Fe oxide-rich deposits. Here, the
establishment of a connection to the Mediterranean Sea through the shallow Bosporus around 9000 years ago
(Degens and Ross, 1974; Soulet et al., 2011) led to the inflow of marine waters into a freshwater basin, resulting in
permanent salinity/density stratification and in the development of euxinic conditions (i.e. free dissolved sulfide
present in the bottom water), making the current Black Sea the largest permanently anoxic basin on Earth.
In the absence of oxygen and metal oxides, $SO_4^{2-}$ reduction is the dominant benthic mineralization process of organic
matter in Black Sea surface sediments below the chemocline  (~ 100 m depth) (Jørgensen et al., 2001; Thamdrup et
al., 2000). At present, $SO_4^{2-}$ penetrates through the modern coccolith ooze (Unit I) and the marine sapropel (Unit II)
sediments and a few meters into the Upper Pleistocene freshwater deposits (Unit III) (Arthur and Dean, 1998;
Degens and Ross, 1974; Jørgensen et al., 2004). Below the $SO_4^{2-}$-bearing zone, methanogenesis takes over as the
dominant process of organic matter degradation, resulting in the buildup of $CH_4$ in the pore water at depth.
Interactions between the cycles of sulfur (S) and $CH_4$ in Black Sea sediments have been extensively studied during
recent years (Holmkvist et al., 2011b; Jørgensen et al., 2001, 2004; Knab et al., 2009; Leloup et al., 2007) and AOM
coupled to $SO_4^{2-}$ reduction ($SO_4$-AOM) was found to account for an estimated 7-18 % of total $SO_4^{2-}$ reduction in
these sediments (Jørgensen et al., 2001). The production of sulfide in the sulfate-methane transition zone (SMTZ) as
a result of $SO_4$-AOM represents the main source of pore water sulfide at depth in the sediment. This intensified
production of sulfide drives an enhanced downward diffusive flux of sulfide into the deep limnic deposits of Unit III,



forming a distinct diagenetic sulfidization front recognized as a black band or a series of bands owing to the
conversion of Fe oxides to Fe sulfides (Jørgensen et al., 2004; Neretin et al., 2004).
At present, the impact of the downward-migrating sulfidization front on sedimentary P, a key nutrient for marine
phytoplankton, and the potential role of Fe-mediated AOM in the deep limnic deposits remain largely unknown. A
buildup of ferrous Fe ($Fe^{2+}$) in the pore water at depth as found in previous studies (Holmkvist et al., 2011b;
Jørgensen et al., 2004; Knab et al., 2009), could indicate ongoing Fe reduction in the $CH_4$-bearing deep limnic
sediments and thus a potential coupling between AOM and Fe oxide reduction. The sediment records investigated up
to now, however, do not extend deep enough to allow the sedimentary cycling of Fe and related biogeochemical
processes below the sulfidization front to be investigated. In particular, the presence of abundant dissolved $Fe^{2+}$
combined with a potential release of pore water phosphate ($HPO_4^{2-}$) during reductive dissolution of Fe oxides may be
conducive to the formation of reduced Fe(II)-P minerals such as vivianite ($Fe_3(PO_4)_2*8H_2O$) below the sulfidization
front (Egger et al., 2015a; Hsu et al., 2014; März et al., 2008). Post-depositional diagenetic alterations as a result of
downward sulfidization could therefore overprint burial records of P in the Upper Pleistocene deposits.
In this study, we combine detailed geochemical analyses of the sediment and pore water with multicomponent
diagenetic modeling to study the diagenetic alterations below the lake-marine transition at two sites in the western
Black Sea. Focusing on the dynamics of S, Fe and P, we demonstrate that AOM coupled to $SO_4^{2-}$ reduction enhances
the downward sulfidization and associated dissolution of Fe oxides, Fe carbonates and vivianite. Below the
sulfidization front, downward diffusing $HPO_4^{2-}$ precipitates as vivianite by reaction with the abundant dissolved $Fe^{2+}$.
We propose that organoclastic Fe oxide reduction and/or AOM coupled to the reduction of Fe oxides are the key
processes explaining the high concentrations of dissolved $Fe^{2+}$ at depth in the sediment. Trends in total S and P with
depth are significantly altered by the above-mentioned reactions, highlighting that diagenesis may strongly overprint
burial records of these elements below a lake-marine transition.
**2        Materials and methods**
**2.1      Sample collection**
**2.1.1    Gravity core sampling**
Sediment samples were taken at two slope sites in the western Black Sea during a cruise in June 2013 with R/V
Pelagia. Gravity cores containing ~ 7 m of sediment were collected at sites 4 (43°40.6' N, 30°7.5' E; 377 meters
below sea surface (mbss)) and 5 (43°42.6' N, 30°6.1' E; 178 mbss) (Fig. 1), both situated below the current
chemocline (~ 100 m water depth). The core liners were pre-drilled with 2 cm diameter holes in two rows of 10 cm
resolution on opposing sides of the tube, offset by 5 cm and taped prior to coring. Upon recovery, the liners were cut
into 1 m sections, transferred to a temperature-controlled container set at in-situ bottom water temperature (11 °C)
and secured vertically. Subsequently, the taped holes were cut open and a cut-off syringe was inserted horizontally
directly after opening each hole.
From one series of holes, 10 mL of wet sediment was extracted at 20 cm resolution and immediately transferred into
a 65 mL glass bottle filled with saturated NaCl solution for $CH_4$ analysis. The NaCl solution was topped up after



addition of the sample, ensuring that no air bubbles remained. Each bottle was sealed with a black rubber stopper and
a screw cap and was subsequently stored upside-down at room temperature. From the second series of holes, 20 mL
sediment was extracted at 20 cm resolution, sealed with parafilm that was tightly closed with an elastic band, and
directly inserted into a nitrogen ($N_2$)-purged glove box. Subsequently, the sediment was transferred into a 50 mL
centrifuge tube and centrifuged (4500 rpm; 30 min). The supernatant from each centrifuged sample was filtered
through 0.45 µm pore size disposable filters via 20 mL plastic syringes in the glove box and collected in 15 mL
centrifuge tubes. The sediment fraction was stored frozen (-20 °C) for solid phase analysis. Filtered pore water
samples were sub-sampled under $N_2$ for analysis of dissolved $HPO_4^{2-}$, ammonium ($NH_4^+$), dissolved inorganic
carbon (DIC), Fe, manganese (Mn), $SO_4^{2-}$ and sulfide ($\sum H_2S = H_2S + HS^-$) (see section 2.2) Additional samples of 10
mL of sediment were collected at approximately 50 cm resolution and transferred into pre-weighed 15 mL glass vials
to determine porosity from gravimetric water loss.
**2.1.2    Multicore sampling**
To sample the surface sediment, sediment cores (30-60 cm of sediment and at least 10 cm of overlying water) were
recovered using an octopus multicorer (core diameter 10 cm). After recovery, the cores were stoppered at the base
and at the top and immediately transported to a temperature-controlled container (11 °C). One multicore from each
cast was pre-drilled with 2 cm diameter holes in two rows at 10 cm resolution on opposing sides of the tube, offset
by 5 cm, and taped prior to coring. These holes were sampled for $CH_4$ as described for the gravity cores. Another
core was directly inserted into a $N_2$-purged glove box through an airtight hole in the base. A bottom water sample
was collected using a 20 mL plastic syringe and the remaining bottom water was removed with a Tygon tube.
Subsequently, the core was sliced anoxically with decreasing resolution at depth, i.e. 0.5 cm resolution for the first 0-
2 cm, 1 cm resolution between 2-10 cm, 2 cm resolution between 10-20 cm and 4 cm resolution for the rest of the
core (> 20 cm). For each slice a sub-sample was placed in a pre-weighed 15 mL glass vial for water content and solid
phase analysis and stored under $N_2$ in airtight jars at -20 °C. A second sub-sample was transferred to a 50 mL
centrifuge tube and centrifuged (4500 rpm; 30 min). Both the supernatant water from each centrifuged sample and
the bottom water sample were subsequently processed as described for the gravity cores.
Visual alignment of the pore water profiles from the multicores with those of the gravity cores showed that the first ~
20 to 30 cm of sediment was lost during long coring. At site 5, the sediment in the multicore consisted of a gray and
homogeneous turbidite below 1.5 cm depth. The depth for the gravity core at site 5 was thus corrected for the loss of
the marine deposits, which were previously reported to be about 50 cm thick at a site in close proximity to site 5
(43°42.63' N, 30°6.12' E; 181 mbss) (Jørgensen et al., 2004)
**2.2    Pore water subsampling**
A sub-sample of 0.5 mL was immediately transferred into a glass vial containing 1.5 mL of 8 M NaOH solution for
analysis of dissolved sulfide. Sub-samples for total dissolved Fe and Mn, which are assumed to represent Fe(II) and
Mn(II), were acidified with 10 µL 35 % suprapur HCl per mL of sub-sample. Another 1 mL of pore water for $HPO_4^{2-}$
analysis was acidified with 4 µL 5 M HCl. Pore water $SO_4^{2-}$ was analyzed with ion chromatography (IC) in a 10-fold





diluted sample (0.15 mL of pore water with 1.35 mL of de-oxygenated UHQ water). Sub-samples for DIC analysis
(0.5 mL) were collected in glass vials (4.9 mL) to which 4.4 mL of 25 g/L NaCl solution was added, making sure
that no headspace remained. Aliquots of the remaining pore water were used for the measurement of alkalinity
(determined onboard by titrating 1 mL of untreated sub-sample with 0.01 M HCl; results presented in the
Supplementary Information only) and $NH_4^+$. All sub-samples were stored at 4 °C and brought to room temperature
just before analysis. Subsampling for sulfide was performed immediately after filtration and all other subsampling
was performed within 4 hours of core recovery.
Pore water sub-samples for sulfide, $HPO_4^{2-}$ and DIC were directly analyzed onboard using an auto analyzer. Sub-
samples for dissolved Fe and Mn were analyzed onshore by ICP-OES (Perkin Elmer Optima 3000 Inductively
Coupled Plasma - Optimal Emission Spectroscopy). For the analysis of pore water $CH_4$, a volume of 10 mL $N_2$ was
injected into the $CH_4$ serum flasks (while a needle inserted through the septum allowed 10 mL of water to escape) to
create a headspace from which a subsample was collected with a gas-tight syringe. Subsequently, $CH_4$ concentrations
were determined in the home laboratory after injection into a Thermo Finnigan Trace GC gas chromatograph (Flame
Ionization Detector). $\delta^{13}C$-$CH_4$ and $\delta D$-$CH_4$ (D, deuterium) were analyzed by Continuous Flow Isotope Ratio Mass
Spectrometry (CF-IRMS) as described in detail in (Brass and Röckmann, 2010) and (Sapart et al., 2011).
### 2.3  Bulk sediment analysis
Sediment samples were freeze-dried, powdered and ground in an agate mortar in an argon (Ar)-filled glove box and
split into oxic and anoxic fractions. Samples from the oxic fraction were used for total elemental and organic carbon
($C_{org}$) analyses under normal atmospheric conditions, whereas anoxic splits for sediment P and Fe speciation were
kept under an inert, oxygen-free Ar or $N_2$ atmosphere at all times to avoid oxidation artefacts (Kraal and Slomp,
2014; Kraal et al., 2009).
#### 2.3.1  Total elemental composition and organic carbon
A split of ~ 125 mg of freeze-dried sediment was dissolved overnight in 2.5 mL HF (40 %) and 2.5 mL of
$HClO_4$/$HNO_3$ mixture, in a closed Teflon bomb at 90 °C. The acids were then evaporated at 160 °C and the resulting
gel was dissolved overnight in 1 M $HNO_3$ at 90 °C. Total elemental concentrations in the 1 M $HNO_3$ solutions were
determined by ICP-OES. A second split of 0.3 g freeze-dried sediment was used to determine the $C_{org}$ content using
an elemental analyzer (Fison Instruments model NA 1500 NCS) after carbonate removal from the sediment with two
washes with 1 M HCl (4 h and 12 h) followed by two washes with UHQ water and subsequent drying of the samples
(Van Santvoort et al., 2002).
#### 2.3.2  Sediment P fractionation
To determine the solid phase partitioning of P, aliquots of 0.1 g dried sediment were subjected to the SEDEX
sequential extraction procedure after Ruttenberg (1992), as modified by Slomp et al. (1996b), but including the first
$MgCl_2$ step (Table 1). Sediment P was fractionated as follows: i) exchangeable-P ("$P_{exch}$", extracted by 1 M $MgCl_2$,
pH 8, 0.5 h), ii) Fe-associated P ("$P_{Fe}$", extracted by citrate-bicarbonate-dithionite (CDB), buffered to pH 7.5 with Na





citrate/Na bicarbonate, 8 h, followed by 1 M $MgCl_2$, pH 8, 0.5 h), iii) authigenic Ca-P ("$P_{authi\ Ca-P}$", including
carbonate fluorapatite, biogenic hydroxyapatite and $CaCO_3$-bound P, extracted by 1 M Na acetate solution, buffered
to pH 4 with acetic acid, 6 h, followed by 1 M $MgCl_2$, pH 8, 0.5 h), iv) detrital Ca-P ("$P_{detr}$", extracted by 1 M HCl,
24 h) and v) organic P ("$P_{org}$", after ashing at 550 °C for 2 h, extracted by 1 M HCl, 24 h). The $MgCl_2$ washes in
steps ii and iii were to ensure that any $HPO_4^{2-}$ re-adsorbed during CDB or acetate extraction was removed and
included in the pools of Fe-associated P and authigenic Ca-P, respectively. Sediments were shielded from oxygen
inside an Ar-filled glovebox until step 3 of the SEDEX procedure to eliminate the potential conversion of Ca-P to
Fe-bound P due to pyrite oxidation upon oxygen exposure (Kraal and Slomp, 2014; Kraal et al., 2009). Dissolved
$HPO_4^{2-}$ in the CDB solution was analyzed by ICP-OES. For all other solutions, $HPO_4^{2-}$ was determined
colorimetrically (Strickland and Parsons, 1972) on a Shimadzu spectrophotometer using the ammonium
heptamolybdate – ascorbic acid method.

### 189    2.3.3    Sediment Fe fractionation

Sediment Fe was fractionated into i) carbonate associated Fe ("$Fe_{carb}$", including siderite and ankerite, extracted by 1
M Na-acetate brought to pH 4.5 with acetic acid, 24 h), ii) easily reducible (amorphous) oxides ("$Fe_{ox1}$", including
ferrihydrite and lepidocrocite, extracted by 1 M hydroxylamine-HCl, 24 h), iii) reducible (crystalline) oxides
("$Fe_{ox2}$", including goethite, hematite and akagenéite, extracted by Na-dithionite buffer, pH 4.8, 2 h) and iv) Fe in
recalcitrant oxides (mostly magnetite, "$Fe_{mag}$", extracted by 0.2 M ammonium oxalate / 0.17 M oxalic acid solution,
2 h), according to Poulton and Canfield (2005), using a 50 mg aliquot of dried sediment (Table 1). An additional
aliquot of 50 mg was subjected to an adapted sequential extraction procedure after Claff et al. (2010), separating
labile Fe(II) ("$Fe(II)_{HCl}$") and Fe(III) ("$Fe(III)_{HCl}$") using 1 M HCl (4 h) from crystalline Fe oxide minerals
("$Fe(II)_{CDB}$", Na-dithionite buffer, pH 4.8, 4 h) and from pyrite ("$Fe_{pyrite}$", concentrated nitric acid, 2 h), for all
multicores as well as for the long core at site 4 (Table 1).
At site 4 (multicore only) and 5 (multicore and gravity core), aliquots of 0.5 g dried sediment were used to
sequentially determine the amount of FeS (acid volatile sulfur, "AVS", using 6 M HCl) and $FeS_2$ (chromium
reducible sulfur, "CRS", using acidic chromous chloride solution) via the passive diffusion method described by
(Burton et al., 2008) using iodometric titration of the ZnS formed in the alkaline Zn acetate traps to quantify AVS
and CRS (Table 1).

### 205    2.4    Diagenetic model

### 206    2.4.1    General form

A multicomponent transient diagenetic model was developed for site 4 based on existing diagenetic models (Reed et
al., 2011a, 2011b; Rooze et al., 2016) to gain a better understanding of the transient diagenesis in Black Sea
sediments and to investigate the potential for Fe-AOM as a source of pore water $Fe^{2+}$ at depth. The model describes
the cycling of dissolved and particulate chemical species in a 1D sediment column (Berner, 1980) and its domain is
represented by 2000 grid cells that capture the upper 2000 cm of the sediment (i.e. vertical resolution of 1 cm). A
total of 25 different chemical species (Table 2) were subjected to a suite of biogeochemical reactions (Table 3) and





vertical transport through burial, as well as molecular diffusion for dissolved species (Boudreau, 1997; Soetaert et
al., 1996; Wang and Van Cappellen, 1996). The general diagenetic equations for solid (Eq. (1)) and dissolved species
(Eq. (2)) are, respectively,
$(1 - \phi)\frac{\partial C_S}{\partial t} = (1 - \phi)v\frac{\partial C_S}{\partial x} + \sum R_S$                  (1)
$\phi\frac{\partial C_{aq}}{\partial t} = \phi D'\frac{\partial^2 C_{aq}}{\partial x^2} - \phi u\frac{\partial C_{aq}}{\partial x} + \sum R_{aq}$                  (2)
where $C_S$ is the concentration of the solid species (mol L$^{-1}$; mass per unit volume of solids), $C_{aq}$ the concentration of
the dissolved species (mol L$^{-1}$; mass per unit volume of pore water), t is time (yr), $\phi$ the sediment porosity, x the
distance from the sediment-water interface (cm), $D'$ the diffusion coefficients of dissolved species in the sediment
(cm$^2$ yr$^{-1}$) adjusted for the considered setting (Supplementary Table S1) (Boudreau, 1997) and corrected for the
tortuosity in the porous medium (Boudreau, 1996) (see Supplementary Information). $\sum R_S$ and $\sum R_{aq}$ are the net
reaction rates of the solid and dissolved species from the chemical reactions they participate in (Table 3), and $v$ and
$u$ the advective velocities (cm yr$^{-1}$) of the solid and the dissolved species, respectively. Porosity and advective
velocities were described by depth-dependent functions to account for sediment compaction (Meysman et al., 2005;
Reed et al., 2011a) (see Supplementary Information and Supplementary Fig. S1).
Reactions considered by the model and corresponding reaction equations are given in Tables 3 and 4, respectively,
and are divided into primary redox reactions and other biogeochemical reactions, including various mineral
formation and dissolution reactions (Reed et al., 2011a, 2011b; Rooze et al., 2016). Corresponding reaction
parameters were mostly taken from the literature or, if these were not available or no fit to the data could be obtained
with existing parameter ranges, constrained using the extensive geochemical dataset for site 4 (Table 5). To account
for differences in reactivity and crystallinity between different species, organic matter and Fe oxides are divided into
three different pools, representing highly reactive (α), less reactive (β) and non-reactive (i.e. inert) (γ) phases. For the
Fe oxides, only the α phase is used by organoclastic Fe reduction (Table 3), while both the α and β phase are used by
Fe-AOM (Rooze et al., 2016).
The succession of oxidants during organic matter decomposition (Froelich et al., 1979) is described by means of
Monod kinetics (Table 4), inhibiting degradation pathways in the presence of oxidants with higher metabolic free
energy yields and switching off pathways when an oxidant is exhausted (Berg et al., 2003; Boudreau, 1996; Reed et
al., 2011b; Rooze et al., 2016; Wang and Van Cappellen, 1996). Corresponding limiting concentrations for the
oxidants are taken from (Reed et al., 2011a) (Table 5). In addition, an attenuation factor, Ψ, is used to slow down
organic matter degradation through $SO_4^{2-}$ reduction and methanogenesis, thus allowing for better preservation of
organic matter under anoxic bottom water conditions (Moodley et al., 2005; Reed et al., 2011a, 2011b).
Cycling of S is simulated using five different chemical species, i.e. Fe monosulfides (FeS), pyrite (FeS$_2$), elemental S
(S$_0$), dissolved sulfide and pore water $SO_4^{2-}$ (Table 2), combined in a network of various biogeochemical reactions
(Table 3). The CH$_4$ cycle includes CH$_4$ production from organic matter and from CO$_2$, as well as CH$_4$ oxidation
coupled to the reduction of O$_2$, $SO_4^{2-}$ and Fe(OH)$_3$ (Table 3). Although Mn-oxides have also been suggested to be a
thermodynamically favorable electron acceptor for AOM (Beal et al., 2009), they were not included in the model
because of the relatively low Mn concentrations (~ 15 μmol g$^{-1}$ for total sedimentary Mn and < 30 μM for dissolved





$Mn^{2+}$; Supplementary Fig. S2 and S3) when compared to Fe and the likely presence of most of the Mn in the form of
Mn-carbonates.
The P forms included in the model are pore water $HPO_4^{2-}$, authigenic Ca-P, organic P and detrital P, as well as Fe-
bound P, i.e. P associated with Fe oxides and P in vivianite (Table 2). The removal of dissolved $Fe^{2+}$ through
formation of the Fe minerals FeS, siderite ($FeCO_3$) and vivianite is also included in the model (Table 3).
The boundary conditions at the sediment surface were specified as time-dependent depositional fluxes for the
particulate components and as fixed bottom water concentrations for the dissolved species, while a zero gradient
boundary condition was set for all chemical species at the base of the model domain (Fig. 2 and Supplementary
Table S2). To avoid potential interferences of the lower boundary conditions with the model results in the upper
sediments, the model depth was set to 2000 cm (see Supplementary Fig. S4). In this paper, only the upper 800 cm are
shown. However, all profiles extending over the full depth range are provided in the Supplementary Information file
(Supplementary Fig. S3 and Fig. S5). The model code was written in R using the marelac geochemical dataset
package (Soetaert et al., 2010) and the ReacTran package (Soetaert and Meysman, 2012) to calculate the transport in
porous media. The set of ordinary differential equations was subsequently solved numerically with the lsoda
integrator algorithm (Hindmarsh, 1983; Petzoldt, 1983)
**2.4.2    Transient scenario**
The model applied in this study simulates the sediment deposition during the last 25000 years. A constant mass
accumulation rate of 0.06 g cm$^{-2}$ yr$^{-1}$ over the Holocene was assumed. In order to reduce the computing time for the
freshwater period, a higher mass accumulation rate of 1 g cm$^{-2}$ yr$^{-1}$ was used between 25000 and 10000 years before
present (B.P.) and all fluxes were corrected accordingly (i.e. multiplied with a factor of 16.67). The best fit to the
chloride (Cl$^-$) profile, which can be used to estimate the timing of the Mediterranean saltwater inflow into the Black
Sea basin, was obtained assuming an initial salinity of 1 for the freshwater lake and a linear increase to a salinity of
22 starting around 9000 years ago (Fig. 2). Such a salinization scenario compares well to a previous salinity
reconstruction by Soulet et al. (2010). However, a constant salinity over the last 2000 years, as suggested by these
authors, resulted in a pore water gradient that was too shallow when compared to the measured pore water Cl$^-$ profile
(Supplementary Fig. S4). Therefore, the period with constant salinity of 22 was adjusted to 100 years to fit the data.
A shift from oxic towards euxinic conditions around 7600 years B.P., with a peak in organic matter loading around
5300 years B.P. and constant elevated organic matter fluxes after 2700 years B.P. was assumed, following a recent
study comprising data from seven sediment cores collected from the Black Sea (Eckert et al., 2013) (Fig. 2). In
addition, the input of organic matter was assumed to increase again in the last century, reflecting anthropogenic
eutrophication of waters on the adjacent continental shelf as previously reported (Capet et al., 2013; Kemp et al.,
2009). With the development of anoxic and sulfidic bottom-water conditions, depositional fluxes of reactive Fe
oxides were assumed to be zero (Fig. 2). In contrast, fluxes of Fe sulfides are high under euxinic conditions and
dominated by $FeS_2$.



## 3 Results

### 3.1 Pore water profiles

Pore water profiles of $SO_4^{2-}$ show a linear decrease from ~ 17 mM at the sediment water interface to a depth of ~ 230 cm at both sites, below which $CH_4$ starts to accumulate in the pore water (Fig. 3). Bubble formation and degassing of $CH_4$ during gravity coring could not be avoided because of the high concentrations of $CH_4$ in the limnic deposits (above the saturation of ca 1.3 mM $CH_4$ at atmospheric pressure; (Jørgensen et al., 2001; Yamamoto et al., 1976)). Higher concentrations measured at site 5 are indicative of less $CH_4$ degassing. Observations of increased bubble formation with depth during coring suggest that decreasing $CH_4$ concentrations below 300 cm reflect enhanced outgassing with increasing levels of $CH_4$ in the deeper sediments. Pore water profiles of $NH_4^+$ at both sites are similar and concentrations increase to ~ 3 mM at depth, suggesting that actual $CH_4$ concentrations at both sites could be comparable. Most of the $CH_4$ values thus only indicate the presence or absence of $CH_4$ and thus are not a quantitative measure. Modeled pore water concentrations of $CH_4$ on the other hand, show a steep increase below the SMTZ, comparable to the gradient observed at site 5, and build up to concentrations of ~ 20 mM at depth (Supplementary Fig. S3).

The removal of both $SO_4^{2-}$ and $CH_4$ around 230 cm depth marks the SMTZ, where $SO_4$-AOM drives the production of dissolved sulfide, DIC and alkalinity (Supplementary Fig. S3) and diffusion of these pore water constituents away from the SMTZ (Fig. 3). Below the sulfide diffusion front, $Fe^{2+}$ accumulates in the pore water. Dissolved $HPO_4^{2-}$ reaches a maximum around the depth where sulfide levels drop below the detection limit of 1 $\mu$mol $L^{-1}$, followed by a steep decrease with depth. Concentrations of pore water $Mn^{2+}$ are more than an order of magnitude lower than those of dissolved $Fe^{2+}$, and decrease from the sediment surface until ~ 200 cm depth, below which they slightly increase again (Supplementary Fig. S3).

The isotopic composition of pore water $CH_4$ (available for site 5 only) seems not affected by the $CH_4$ loss and reveals a biological origin in the limnic deposits, with hydrogenotrophic carbonate reduction, i.e. microbial reduction of $CO_2$ to $CH_4$ as the main methanogenic pathway for the range of $CH_4$ isotope ratios observed in these sediments (Fig. 4) (Whiticar, 1999). Upward diffusing $CH_4$ shows a gradual depletion in $\delta^{13}C$-$CH_4$ from ~ -74 ‰ at depth to ~ -96 ‰ around the SMTZ, followed by subsequent progressive $^{13}C$ enrichment towards the sediment surface. $\delta D$-$CH_4$ shows a small enrichment from -226 ‰ at depth to ~ -208 ‰ at the SMTZ and a strong shift towards high $\delta D$-$CH_4$ values of up to ~ 113 ‰.

### 3.2 Solid phase profiles

A pronounced excursion in sedimentary $C_{org}$ at site 4 in combination with a shift from gray clay deposits to micro-laminated black sediments indicates that the lake-marine transition, i.e. the transition between the marine sapropel Unit II and the deep limnic sediments of Unit III (Arthur and Dean, 1998; Degens and Ross, 1974), is located around a sediment depth of ~ 90 cm at site 4 (Fig. 5). At site 5, Unit I and Unit II were lost due to a turbidite, explaining the low concentrations of $C_{org}$ in the upper sediments.

Concentrations of solid S increase with decreasing depth from 20 $\mu$mol $g^{-1}$ below 300 cm (sulfidization front) to ~ 400 $\mu$mol $g^{-1}$ in the upper 100 cm at both sites and are dominated by $FeS_2$ (Fig. 5). Iron oxides show a decrease from



~ 100 µmol g$^{-1}$ at depth to ~ 50 µmol g$^{-1}$ in the sediments between 100 – 300 cm and a further decrease to ~ 10 µmol
g$^{-1}$ closer to the sediment surface. Amorphous Fe oxides (Fe$_{ox1}$) and more crystalline oxides (Fe$_{ox2}$) both account for
half the total amount of Fe oxides, with a small contribution of recalcitrant oxides (Fe$_{mag}$) (Supplementary Fig. S2).
The results from the two different Fe extractions applied in this study (Table 1) generally compare well
(Supplementary Fig. S2). Note, however, that the Fe oxides in Fig. 5 represent the results from the extraction after
Poulton and Canfield (2005). Results from the Fe extractions modified from Claff et al. (2010) are provided in the
Supplementary Information only. Sedimentary Mn content is relatively low at all three sites, ranging from ~ 5-10
µmol g$^{-1}$ in the marine sediments to ~ 15 µmol g$^{-1}$ in the deep limnic deposits of Unit III (Supplementary Fig. S2).
Sediments below the sulfidization front are characterized by high Fe carbonate contents of ~ 100 µmol g$^{-1}$. The sharp
depletion in Fe carbonate around the sulfidization front could only be reproduced in the model by assuming Fe
carbonate dissolution by dissolved sulfide (Table 3). These results suggest a conversion of reactive Fe from
carbonate toward sulfide phases in the presence of abundant dissolved sulfide.
Units I and II show high concentrations of organic P, which accounts for ~ 30 % of total P in these sediments (Fig.
5). Low organic P and high concentrations of detrital P in the upper sediments at site 5 are due to the turbidite. The
limnic deposits of Unit III are generally depleted in organic P (< 6 % of total P) and enriched in detrital P.
Authigenic Ca-P shows little variation in the sediments of Unit III, accounting for ~ 20 to 30 % of total P at the two
sites. The contribution of Fe-associated P, on the other hand, is reduced in the limnic deposits of Unit III exposed to
the downward diffusing sulfide (~ 20 %) when compared to the sediments below the sulfidization front (~ 30 %).
Concentrations of exchangeable P are < 2 µmol g$^{-1}$ for sediments above the SMTZ and < 1 µmol g$^{-1}$ for sediments at
depth (data not shown).
Modeled SO$_4^{2-}$ reduction rates show two distinct peaks of ~ 2 nmol cm$^{-3}$ d$^{-1}$ in the sediments of Unit I and II, as well
as an additional peak in the sediments around the SMTZ (Fig. 6). Rates of methanogenesis are highest in the organic-
rich marine deposits (~ 0.2 - 0.3 nmol cm$^{-3}$ d$^{-1}$) and generally around ~ 50 pmol cm$^{-3}$ d$^{-1}$ in the limnic deposits. The
sediments around the SMTZ are further characterized by high rates of SO$_4$-AOM (~ 0.3 nmol cm$^{-3}$ d$^{-1}$), whereas
sediments directly below the sulfidization front show enhanced rates of S$_0$ disproportionation (~ 60 pmol cm$^{-3}$ d$^{-1}$).
Organoclastic SO$_4^{2-}$ reduction provides the main source for pore water sulfide in the organic-rich marine deposits,
while SO$_4$-AOM and S$_0$ disproportionation are the dominant sources of dissolved sulfide in sediments around the
SMTZ and directly below the sulfidization front, respectively. Rates of Fe-AOM are generally low (~ 0.1 pmol cm$^{-3}$
d$^{-1}$) and restricted to the limnic deposits only.
The temporal evolution in pore water and solid phase constituents illustrates the impact of the lake-marine transition
on the sediment geochemistry (Fig. 7). Concentrations of pore water Cl$^-$ and SO$_4^{2-}$ increase with the intrusion of
marine Mediterranean Sea waters ca. 9000 years ago, accompanied by a decrease in dissolved CH$_4$ and accumulation
of pore water sulfide in the shallower sediments. Dissolved Fe$^{2+}$ becomes restricted to non-sulfidic pore waters at
depth, while HPO$_4^{2-}$ and solid S start to accumulate in the presence of dissolved sulfide. Iron oxides decrease in the
surface sediments as well as in the sediments at depth. Vivianite, on the other hand, becomes increasingly enriched in
sediments below the downward diffusing sulfide front.





**4.     Discussion**
**4.1 Coupled S, CH$_4$ and Fe dynamics**
**4.1.1 Organoclastic SO$_4^{2-}$ reduction**
Model-derived areal rates of SO$_4^{2-}$ reduction of ~ 0.72 mmol m$^{-2}$ d$^{-1}$ (Table 6), i.e. the total amount of SO$_4^{2-}$ reduced
per square meter of sea floor, are in good agreement with previous estimates of 0.65-1.43 mmol m$^{-2}$ d$^{-1}$ for sediments
of the Black Sea (Jørgensen et al., 2001). SO$_4^{2-}$ reduction accounts for > 90 % of total organic matter degradation in
the model (Table 6), supporting previous conclusions that SO$_4^{2-}$ reduction represents the dominant mineralization
process of organic matter in sediments below the chemocline (Jørgensen et al., 2001; Thamdrup et al., 2000).
The depth-dependent rate profile of SO$_4^{2-}$ reduction shows two distinct peaks of ~ 2 nmol cm$^{-3}$ d$^{-1}$ associated with
organoclastic SO$_4^{2-}$ reduction in the organic matter rich marine deposits of Unit I and Unit II. These high rates
compare well with literature values of 0.1 - 20 nmol cm$^{-3}$ d$^{-1}$ (Holmkvist et al., 2011b; Jørgensen et al., 2001, 2004;
Knab et al., 2009; Leloup et al., 2007). Thus, like previous modeling approaches based on hybrid modeling with
experimentally measured SO$_4^{2-}$ reduction rates (SRR) in the uppermost sediment layers (Jørgensen et al., 2001), the
transient diagenetic model developed in this study is capable of reproducing the high rates of SO$_4^{2-}$ reduction near the
sediment surface. Our model further demonstrates that the two SRR peaks in the sediments of Unit I and Unit II are
not reflected in the pore water profile of SO$_4^{2-}$, indicating that SRR estimates based on pore water profiles of SO$_4^{2-}$
alone may underestimate the actual rate of SO$_4^{2-}$ reduction in marine sediments.
**4.1.2 SO$_4$-AOM**
Pore water profiles of SO$_4^{2-}$, CH$_4$, sulfide and DIC reveal a distinct SMTZ around 230 cm depth at both sites, where
SO$_4$-AOM with upward diffusing CH$_4$ results in the concomitant removal of pore water SO$_4^{2-}$ and CH$_4$ and in the
accumulation of dissolved sulfide and DIC in the pore waters of these sediments (Fig. 3). The depth of the SMTZ
and the steep increase in CH$_4$ to > 3 mM below the SMTZ found in this study are consistent with earlier observations
in sediments of the western Black Sea (Holmkvist et al., 2011b; Jørgensen et al., 2001, 2004; Knab et al., 2009;
Leloup et al., 2007). The location of the SMTZ, however, has progressed downwards in the last 9000 years,
following the inflow of SO$_4^{2-}$-rich salt water into the Black Sea basin (Fig. 7).
In the model, SO$_4$-AOM results in enhanced rates of SO$_4^{2-}$ reduction at the SMTZ of ~ 0.3 nmol cm$^{-3}$ d$^{-1}$ (Fig. 8).
Calculated areal rates of SO$_4$-AOM of ~ 0.17 mmol m$^{-2}$ d$^{-1}$ suggest that AOM accounts for ~ 19 % of the total SO$_4^{2-}$
reduction in these sediments (Table 6). Such a high contribution of AOM is close to the range of previous estimates
of 7-18 % (Jørgensen et al., 2001, 2004). Around the SMTZ, SO$_4$-AOM is responsible for ~ 90 % of the total SO$_4^{2-}$
reduction (Fig. 6 and Table 6), thus enhancing the downward diffusive flux of sulfide into the deep limnic deposits of
Unit III. Our model suggests that without this additional source of sulfide through SO$_4$-AOM, the sulfidization front
would currently be located around 150 cm depth in the sediment (Fig. 8).
The consumption of upward diffusing CH$_4$ by SO$_4^{2-}$-driven AOM leads to a progressive enrichment of $^{13}$C and D in
the residual CH$_4$ above the SMTZ (Fig. 4) due to the preferential oxidation of isotopically light CH$_4$ during SO$_4$-
AOM (Alperin et al., 1988; Martens et al., 1999; Whiticar, 1999). Modeled concentrations of CH$_4$ indicate that the



measurements above the sulfidization front at site 5 are likely less affected by outgassing during core recovery (Fig.
4) and can thus be used to derive kinetic isotope fractionation factors for carbon ($\varepsilon_C$) and hydrogen ($\varepsilon_H$) associated
with $SO_4$-AOM at the SMTZ using the Rayleigh distillation function (Crowe et al., 2011; Egger et al., 2015b;
Rayleigh, 1896; Whiticar, 1999). Corresponding estimates for $\varepsilon_C$ of ~ 8 ‰ ($R^2$ = 0.972) and $\varepsilon_H$ of ~ 58 ‰ ($R^2$ =
0.982) are at the lower end of previously documented values in marine and brackish-marine environments (8-38 ‰
for $\varepsilon_C$ and 100-324 ‰ for $\varepsilon_H$) (Alperin et al., 1988; Egger et al., 2015b; Holler et al., 2009; Martens et al., 1999;
Reeburgh, 2007). At the base of the SMTZ, however, upward diffusing $CH_4$ reveals an initial depletion in $\delta^{13}C$-$CH_4$
(Fig. 4). Such a shift to $^{13}C$-depleted $CH_4$ together with a decrease in its concentration could indicate an enzyme-
mediated equilibrium C isotope exchange during $SO_4$-AOM at low $SO_4^{2-}$ concentrations (< 0.5 mM) (Holler et al.,
2012; Yoshinaga et al., 2014). The effect of such mechanisms on deuterated $CH_4$ is likely limited.

### 4.1.3 Cryptic S cycling

Earlier studies postulated ongoing $SO_4^{2-}$ reduction (< 1 nmol $cm^{-3}$ $d^{-1}$) within the $SO_4^{2-}$-depleted (< 0.5 mM) limnic
deposits below the SMTZ in sediments of the Black Sea (Holmkvist et al., 2011b; Knab et al., 2009; Leloup et al.,
2007), Baltic Sea (Holmkvist et al., 2011a, 2014; Leloup et al., 2009) and Alaskan Beaufort Sea (Treude et al., 2014)
likely driven by $SO_4^{2-}$ production from re-oxidation of dissolved sulfide with oxidized Fe minerals. In this
mechanism, Fe oxides enhance the recycling of sulfide to $SO_4^{2-}$ in a cryptic S cycle (Holmkvist et al., 2011a; Treude
et al., 2014) thereby fueling $SO_4^{2-}$-driven AOM in Fe oxide-rich sediments. In this cryptic S cycle, dissolved sulfide
is oxidized to zero-valent sulfur ($S_0$), a key intermediate in AOM, which is subsequently disproportionated to $SO_4^{2-}$
and sulfide by associated Deltaproteobacteria (Holmkvist et al., 2011a; Milucka et al., 2012; Sivan et al., 2014;
Treude et al., 2014). The additional $SO_4^{2-}$, produced during $S_0$ disproportionation, may then be re-used by the
methanotrophic archaea as an electron acceptor for $SO_4$-AOM (Milucka et al., 2012).
Our model results suggest slow rates of ongoing $SO_4^{2-}$ reduction of < 0.2 nmol $cm^{-3}$ $d^{-1}$ (Fig. 6) within the limnic
deposits exposed to dissolved sulfide (Table 6), in line with estimated SRR based on $^{35}SO_4^{2-}$ incubation experiments
with Black Sea sediments from below the SMTZ of ~ 0.1-0.5 nmol $cm^{-3}$ $d^{-1}$ (Knab et al., 2009; Leloup et al., 2007).
Below the sulfidization front, SRR drop to ~ 2 pmol $cm^{-3}$ $d^{-1}$, but remain above zero. Active $SO_4^{2-}$ reduction in these
$SO_4^{2-}$-depleted sediments requires deep $SO_4^{2-}$ formation to maintain low net rates of $SO_4^{2-}$ reduction. In the model, $S_0$
disproportionation is the only potential source of pore water $SO_4^{2-}$ at depth (Table 3). Formation of $S_0$, in turn, occurs
exclusively by oxidation of dissolved sulfide during the reductive dissolution of Fe oxides, explaining the distinct $S_0$
disproportionation peak of ~ 60 pmol $cm^{-3}$ $d^{-1}$ around the sulfidization front (Fig. 6). Thus, based on the model
assumptions, we conclude that Fe oxides increase the transformation of sulfide to $SO_4^{2-}$ via formation and subsequent
disproportionation of $S_0$ in these sediments, as suggested previously (Holmkvist et al., 2011b; Knab et al., 2009;
Leloup et al., 2007). Such recycling of $SO_4^{2-}$ stimulates slow rates of $SO_4$-AOM in the sediments below the SMTZ,
explaining the low background rates of $SO_4^{2-}$ reduction throughout the limnic deposits at depth (~ 2 pmol $cm^{-3}$ $d^{-1}$).
These results support recent findings of indirect Fe stimulated $SO_4^{2-}$ driven AOM in laboratory experiments (Sivan et
al., 2014), and highlight that Fe oxides could play a significant role as stimulators of AOM and S recycling in natural
environments.



### 4.2 Fe reduction below the sulfidization front

Below the sulfidization front, $Fe^{2+}$ starts to accumulate in the pore water (Fig. 3). Although previous studies have also reported an increase of dissolved $Fe^{2+}$ around the depth where sulfide levels drop below the detection limit (Holmkvist et al., 2011b; Jørgensen et al., 2004; Knab et al., 2009), the source of this pore water $Fe^{2+}$ has remained unknown. One possible explanation could be that the elevated $Fe^{2+}$ concentrations at depth represent remnant $Fe^{2+}$ accumulated during the Black Sea "Lake" phase (Knab et al., 2009). In our model, $Fe^{2+}$ shows a broad peak of ~ 300 µM until ~ 300 cm depth in the sediment during the initial Lake phase, assuming organoclastic Fe reduction as the only Fe reduction pathway (data not shown). The removal of $Fe^{2+}$ through authigenic formation of reduced Fe(II) minerals, however, prevents the accumulation of substantial amounts of $Fe^{2+}$ in the pore water below ~ 300 cm sediment depth during the Lake phase (Fig. 8). We therefore conclude that the high concentrations of dissolved $Fe^{2+}$ below the sulfidization front are most likely indicative of active Fe reduction in these sediments.

### 4.2.1 Fe reduction through cryptic S cycling

In theory, a cryptic S cycle, as described in section 4.1.3, could result in net accumulation of dissolved $Fe^{2+}$ if the sulfide consumption from reaction with ferric Fe outweighs the production of sulfide from $SO_4^{2-}$ reduction. Modeled $Fe^{2+}$ indeed shows a peak of < 100 µM directly below the sulfidization front, assuming no active Fe reduction in the limnic deposits (Fig. 8). However, concentrations of dissolved $Fe^{2+}$ are too low compared to the measurements and confined to sediments between 300 – 400 cm depths only. The diagenetic model developed in this study therefore suggests that cryptic S cycling cannot explain the high concentrations of dissolved $Fe^{2+}$ in the deep limnic deposits.

### 4.2.2 Organoclastic Fe reduction

In the model, the reduction of Fe oxides coupled to organic matter degradation only occurs with the easily reducible α phase in order to allow for the burial of the more crystalline β phase at depth (Table 3) (Rooze et al., 2016). Since the α phase is efficiently reduced in the upper few centimeters during organoclastic Fe reduction, no easily reducible Fe oxides are being buried into the deep sediments in the diagenetic model. Organoclastic Fe reduction therefore does not occur within the modeled deep limnic deposits that exclusively contain more crystalline (β) and refractory (γ) Fe oxides (Fig. 5). Instead, we assume that $CH_4$ represents a plausible electron donor for the reduction of more crystalline Fe oxides in the organic-poor deep sediments with relatively refractory old organic matter (< 0.8 wt %). This assumption is supported by an increasing body of geochemical evidence and laboratory incubation experiments showing that Fe-AOM might be occurring in a variety of different aquatic environments (Amos et al., 2012; Beal et al., 2009; Crowe et al., 2011; Egger et al., 2015b; Riedinger et al., 2014; Scheller et al., 2016; Segarra et al., 2013; Sivan et al., 2011; Wankel et al., 2012).

In addition, several studies have shown that Fe-reducing microorganisms are able to outcompete methanogens for common substrates (e.g. acetate and $H_2$), thus reducing the concentrations of these common primary electron donors to levels that are too low for methanogens to grow (Achtnich et al., 1995; Lovley and Phillips, 1987; Lovley et al., 1989). These results, together with the observed capability of methanogens to switch from $CH_4$ production to Fe reduction (Bodegom et al., 2004; Bond and Lovley, 2002; Liu et al., 2011; Reiche et al., 2008; Sivan et al., 2016;



Vargas et al., 1998) led to the common conclusion that Fe oxides exert a suppressive effect on methanogenesis.
Ongoing $CH_4$ production in the Fe oxide-rich limnic deposits, as deduced from the isotopic composition of pore
water $CH_4$ (Fig. 4) could then indicate limited organoclastic Fe reduction in these sediments.
However, there is increasing evidence that (semi)conductive crystalline Fe oxides (e.g. hematite and magnetite) can,
in fact, stimulate concurrent methanogenesis and organoclastic Fe reduction through direct interspecies electron
transfer (DIET), by serving as electron conduits among syntrophic $CH_4$-producing organisms at rates that are
substantially higher than those for interspecies electron transfer by $H_2$ (Cruz Viggi et al., 2014; Kato et al., 2012; Li
et al., 2014; Zhou et al., 2014; Zhuang et al., 2015). The inhibitory effect of Fe reduction on methanogenesis thus
appears to be lower for crystalline Fe oxides such as hematite and magnetite, which are less bioavailable to Fe-
reducing organisms than poorly crystalline (amorphous) Fe oxides (e.g. ferrihydrite and lepidocrocite) (Lovley,
1991; Qu et al., 2004; Zhuang et al., 2015). These findings indicate that the crystallinity and conductivity of Fe
oxides may play a key role in determining whether methanogenesis is stimulated or suppressed in Fe oxide-rich
environments.
The presence of methanogens that are able to rapidly switch between methanogenesis and reduction of Fe oxides
could also result in a reactivation of less reactive Fe oxides that were not reduced during initial organoclastic Fe
reduction in the deep methanogenic zone as suggested by Sivan et al. (2016). Thus, the deep limnic sediments may
be characterized by a complex interplay of concurrent methanogenesis, Fe oxide reduction and methanotrophy, i.e.
AOM.
**4.2.3 Fe-AOM**
Our model results indicate that Fe-AOM could also be a possible mechanism explaining the buildup of pore water
$Fe^{2+}$ below the sulfidization front. Previous studies have shown that in systems where production and oxidation of
$CH_4$ take place concurrently, methanogenesis might conceal the isotopic signature of AOM (Egger et al., 2015b;
Seifert et al., 2006; Whiticar, 1999). Thus, unlike $SO_4$-AOM, Fe-dependent AOM likely only has little effect on the
isotopic composition of pore water $CH_4$ due to the removal of small amounts of $CH_4$ in sediments with ongoing
methanogenesis. This might explain why pore water $CH_4$ does not show enrichment in both heavy isotopes below the
sulfidization front as would be expected if Fe-AOM would occur, but rather indicates antipathetic changes, i.e.
depletion in $^{13}C$-$CH_4$ and enrichment in D-$CH_4$, usually attributed to $CH_4$ production from carbonate reduction
(Chanton et al., 2005; Whiticar, 1999).
Model derived rates for Fe-AOM of ~ 0.1 pmol $cm^{-3}$ $d^{-1}$ (Fig. 6) are significantly lower than potential Fe-AOM rates
of ~ 4 nmol $cm^{-3}$ $d^{-1}$ estimated from laboratory incubation studies (Egger et al., 2015b; Segarra et al., 2013; Sivan et
al., 2011) with brackish and limnic sediment samples. This large deviation is likely due to an overestimation of Fe-
AOM rates derived from stimulated microbial communities under laboratory conditions using freshly synthesized
and thus easily bioavailable Fe oxides when compared to in-situ conditions.
In the upper 800 cm of sediment, Fe-AOM accounts for < 0.1 % of total $CH_4$ oxidation, with the remaining > 99.9 %
attributed to $SO_4$-AOM (Table 6). Below the sulfidization front, Fe-AOM contributes to ~ 10 % of total $CH_4$
removal. However, while high rates of $SO_4$-AOM are mainly restricted to the SMTZ, Fe-AOM might occur over a





deep methanogenic zone, reaching far down into the sediment. To accurately assess the contribution of Fe-AOM to
the total $CH_4$ consumption in Black Sea sediments, additional knowledge about the vertical expansion of the Fe
oxide-rich limnic sediments deposited during the Blake Sea "Lake" phase would be required.

**4.3 Impact of S-Fe-CH₄ dynamics on sedimentary P diagenesis**

Degradation of organic matter and the subsequent release of $HPO_4^{2-}$ to the pore water during early diagenesis
typically results in a sink-switching from organic P to authigenic P-bearing phases such as Ca phosphates (Filippelli,
1997; Ruttenberg and Berner, 1993; Slomp et al., 1996a), Mn-Ca carbonates (Jilbert and Slomp, 2013; Mort et al.,
2010; Suess, 1979) or reduced Fe phosphates (Burns, 1997; Jilbert and Slomp, 2013; Martens et al., 1978; März et
al., 2008). Reductive dissolution of Fe oxides by dissolved sulfide and the following liberation of $HPO_4^{2-}$ may also
contribute to the buildup of pore water $HPO_4^{2-}$ (Burns, 1997; Egger et al., 2015a; März et al., 2008; Schulz et al.,
1994). Thus, the downward sulfidization ultimately results in the accumulation of dissolved $HPO_4^{2-}$ in the pore water
as the sulfidization front moves downward into the limnic deposits (Fig. 7).
The pore water profile of $HPO_4^{2-}$ (Fig. 3) indicates the presence of a sink for $HPO_4^{2-}$ below the sulfidization front
and, to a lesser extent, in the sulfidic sediments around the SMTZ, likely unrelated to Ca-P authigenesis (Fig. 5).
Such a sink for $HPO_4^{2-}$ below sulfidic sediments has been observed previously (Burns, 1997; Egger et al., 2015a;
März et al., 2008; Schulz et al., 1994; Slomp et al., 2013) and shown to be most likely the result of vivianite
formation (Egger et al., 2015a; Hsu et al., 2014; März et al., 2008). Abundant dissolved $Fe^{2+}$ and a peak in Fe-
associated P below the sulfidization front observed in this study (Fig. 3 and Fig. 5) suggest that vivianite authigenesis
might also be occurring in the limnic deposits below the sulfidzation front in Black Sea sediments.
Assuming that vivianite formation represents the only sink for pore water $HPO_4^{2-}$ results in a good fit between the
modeled and measured pore water profile of $HPO_4^{2-}$ below the sulfidization front (Fig. 3). Modeled vivianite
formation accounts for up to 70 % of total Fe-associated P directly below the sulfidization front. However, the model
underestimates the sharp peak in Fe-associated P directly below the sulfidization front, suggesting that modeled
vivianite formation likely underestimates the actual contribution of vivianite in these sediments. In the limnic
deposits not yet impacted by the downward sulfidization, modeled vivianite accounts for ~ 20 − 30 % of total Fe-
associated P. From this, we estimate that vivianite may be responsible for > 20 % of total P burial directly below the
sulfidization front and for ~ 10 % of total P burial in the deep limnic deposits at depth.
Running the model without Fe-AOM and thus without a source of dissolved $Fe^{2+}$ at depth results in a modeled
vertical $HPO_4^{2-}$ pore water profile of ~ 300 μM at depth in the sediment (Fig. 8). This suggests that Fe-AOM can
promote conditions that allow sequestration of a significant proportion of P as vivianite in the limnic deposits below
the sulfidization front. Consistent with earlier findings, Fe-AOM likely only accounts for a small fraction of total
$CH_4$ oxidation, but may substantially impact the biogeochemical cycling of sedimentary P (Egger et al., 2015a,
2015b; Rooze et al., 2016).
The deviation between the modeled and measured profiles of $HPO_4^{2-}$ and Fe-associated P around the SMTZ (Fig. 3
and Fig. 5) could indicate the formation of vivianite in microenvironments as previously suggested for sulfidic
sediments (Dijkstra et al., 2014; Jilbert and Slomp, 2013). For example, *Deltaproteobacteria*, known to be involved





in $SO_4$-AOM, have been shown to accumulate Fe- and P-rich inclusions in their cells (Milucka et al., 2012). They
may therefore provide a potential explanation for the occurrence of Fe-associated P in sulfidic sediments (Dijkstra et
al., 2014; Jilbert and Slomp, 2013). However, such microenvironments are not captured in our model.
In the diagenetic model, vivianite undergoes dissolution if sulfide is present in the pore waters (Table 3). Sulfide-
induced vivianite dissolution significantly improved the model fit to the measured $HPO_4^2$ and sulfide data. With the
downward migration of dissolved sulfide, modeled vivianite becomes increasingly enriched below the sulfidization
front (Fig. 7). Thus, similar to the sulfidization front, a downward diffusive vivianite front may exist in sedimentary
systems experiencing downward sulfidzation.
In summary, the enhanced downward sulfidization driven by $SO_4$-AOM leads to dissolution of Fe oxide-bound P in
the lake deposits. Below the sulfidization front, downward diffusing $HPO_4^{2-}$ is bound again in authigenic vivianite
due to high concentrations of dissolved $Fe^{2+}$ at depth in the sediment generated by ongoing Fe oxide reduction. As a
result, trends in total P with depth are significantly altered, showing an accumulation in total P below the
sulfidization front unrelated to changes in organic matter deposition and enhanced sedimentary P burial during
deposition.
**5.       Conclusions**
In the Black Sea, the shift from a freshwater lake to a marine system and subsequent downward diffusion of marine
$SO_4^{2-}$ into the $CH_4$-bearing lake sediments results in a multitude of diagenetic reactions around the SMTZ (Fig. 9).
The diagenetic model developed in this study shows that $SO_4$-AOM within the SMTZ significantly enhances the
downward diffusive flux of sulfide into the deep limnic deposits, forming a distinct diagenetic sulfidization front
around 300 cm depth in the sediment. Our results indicate that without this additional source of dissolved sulfide in
the SMTZ, the current sulfidization front would be located around a depth of 150 cm. During the downward
sulfidization, Fe oxides, Fe carbonates and vivianite are converted to Fe sulfide phases, leading to an enrichment in
solid phase S contents and the release of $HPO_4^{2-}$ to the pore water. Our results further support the hypothesis that part
of the downward migrating sulfide is re-oxidized to $SO_4^{2-}$ upon reaction with ferric Fe minerals, fueling a cryptic S
cycle and thus stimulating slow rates (~ 2 pmol $cm^{-3}$ $d^{-1}$) of $SO_4$-AOM in the $SO_4^{2-}$-depleted limnic deposits below
the SMTZ (Holmkvist et al., 2011a, 2011b; Knab et al., 2009; Leloup et al., 2007).
We propose that besides organoclastic Fe oxide reduction, AOM coupled to the reduction of Fe oxides may also be a
possible mechanism explaining the high concentrations of $Fe^{2+}$ in the pore water below the sulfidization front. The
buildup of dissolved $Fe^{2+}$ at depth creates conditions that allow sequestration of the downward diffusing $HPO_4^{2-}$ as
authigenic vivianite, resulting in an accumulation of total P in these sediments.
The diagenetic processes described here reveal that AOM may strongly overprint burial records of Fe, S and P in
depositional marine systems subject to changes in organic matter loading or water column salinity such as coastal
environments (Egger et al., 2015a; Rooze et al., 2016), deep-sea fan sediments (März et al., 2008; Schulz et al.,
1994) and many high-latitude seas (Holmkvist et al., 2014; Treude et al., 2014). Interpreting these diagenetic patterns
as primary sedimentary signals may lead to incorrect reconstructions of environmental conditions during sediment
deposition.




**Acknowledgments**
We thank the captain, crew and shipboard party of the PHOXY cruise aboard R/V Pelagia to the Black Sea in June
2013. We also thank NIOZ Marine Research Facilities for their support and K. Bakker and S. Ossebaar for their
contribution to the pore water analysis. D. van de Meent, T. Claessen, T. Zalm, A. van Dijk, E. Dekker and G.
Megens are acknowledged for technical and analytical assistance in Utrecht. We further thank C. van der Veen for
the methane isotope analysis. This research was funded by ERC Starting Grant 278364, NWO Open Competition
Grant 822.01013 and NWO-Vici Grant 865.13.005 (to C. P. Slomp). This work was carried out under the program of
the Netherlands Earth System Science Centre (NESSC), financially supported by the Ministry of Education, Culture
and Science (OCW).

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





**Tables**
**Table 1. Overview of the sequential P, Fe and S fractionation methods used in this study.**

| Step and code | Extractant, extraction time | Target phase |
|---|---|---|
| **P fractionation (modified from Ruttenberg (1992); done for site 4 (MC & GC) and site 5 (MC & GC))** | | |
| **1 $P_{exch}$** | 1 M $MgCl_2$, pH 8, 0.5 h | Exchangeable P |
| **$2^a$ $P_{Fe}$** | 25 g L-1 Na dithionite, pH 7.5, 8 h | Fe-associated P |
| **$3^a$ $P_{authi\ Ca-P}$** | Na acetate buffer, pH 4, 6 h | P in authigenic and biogenic Ca-P minerals and $CaCO_3$ |
| **4 $P_{detr}$** | 1 M HCl, 24 h | Detrital P |
| **5 $P_{org}$** | Ashing at 550 °C (2h), then 1 M HCl, 24 h | Organic P |
| **Fe fractionation (after Poulton and Canfield (2005); done for site 4 (MC & GC) and site 5 (MC))** | | |
| **1 $Fe_{carb}$** | 1 M Na acetate, pH 4.5, 24 h | Carbonate-associated Fe |
| **2 $Fe_{ox1}$** | 1 M hydroxylamine-HCl, 24 h | Amorphous Fe oxides (ferrihydrite) |
| **3 $Fe_{ox2}$** | 50 g L-1 Na dithionite, pH 4.8, 2 h | Crystalline Fe oxides (goethite, hematite) |
| **4 $Fe_{mag}$** | 0.2 M ammonium oxalate/ 0.17 M oxalic acid, 2 h | Recalcitrant Fe oxides (mostly magnetite) |
| **Fe fractionation (modified from Claff et al. (2010); done for site 4 (MC & GC) and site 5 (MC))** | | |
| **1 $Fe(II)_{HCl}$** | 1 M HCl, 4 h | Labile Fe (carbonates, poorly ordered sulfides) |
| **2 $Fe(III)_{HCl}$** | 1 M HCl, 4 h | Labile Fe (easily reducible oxides) |
| **3 $Fe(III)_{CDB}$** | 50 g L-1 Na dithionite, pH 4.8, 4 h | Crystalline Fe oxides |
| **4 $Fe_{pyrite}$** | Concentrated $HNO_3$, 2 h | Pyrite ($FeS_2$) |
| **S fractionation (after Burton et al. (2008); done for site 4 (MC) and site 5 (MC & GC))** | | |
| **1 AVS** | 6 M HCl, 24 h | S in Fe monosulfides (FeS) |
| **2 CRS** | Acidic chromous chloride solution, 48 h | S in pyrite ($FeS_2$) |

[a]These steps were followed by a wash step with 1 M $MgCl_2$, which was added to the corresponding step. MC = multicore and GC
= gravity core.





**Table 2. Chemical species included in the diagenetic model.**

| Species | Notation | Type |
| --- | --- | --- |
| Organic matter[a] | $OM^{\alpha,\beta,\gamma}$ | Solid |
| Iron oxides[a] | $Fe(OH)_3^{\alpha,\beta,\gamma}$ | Solid |
| Iron monosulfide | $FeS$ | Solid |
| Pyrite | $FeS_2$ | Solid |
| Siderite | $FeCO_3$ | Solid |
| Elemental sulfur | $S_0$ | Solid |
| Iron oxide-bound phosphorus | $Fe_{ox}P$ | Solid |
| Vivianite | $Fe_3(PO_4)_2$ | Solid |
| Organic phosphorus | $P_{org}$ | Solid |
| Authigenic (Ca) phosphorus | $CaP$ | Solid |
| Detrital phosphorus | $DetrP$ | Solid |
| Chloride | $Cl^-$ | Solute |
| Oxygen | $O_2$ | Solute |
| Sulfate | $SO_4^{2-}$ | Solute |
| Iron | $Fe^{2+}$ | Solute |
| Hydrogen sulfide[b] | $\sum H_2S$ | Solute |
| Methane | $CH_4$ | Solute |
| Ammonium[b] | $\sum NH_4^+$ | Solute |
| Nitrate | $NO_3^-$ | Solute |
| Phosphate | $\sum HPO_4^{2-}$ | Solute |
| Dissolved inorganic carbon | $DIC$ | Solute |

[a] There are three types of species: reactive (α), less reactive (β) and refractory (γ)
[b] $\sum$ denotes that all species of an acid are included



**Table 3. Reaction pathways and stoichiometries implemented in the diagenetic model.**

| Primary redox reactions* | |
|---|---|
| $OM^{\alpha,\beta} + aO_2 \rightarrow aCO_2 + bNH_4^+ + cH_3PO_4 + aH_2O$ | R1 |
| $OM^{\alpha,\beta} + \frac{4a}{5}NO_3^- + \frac{4a}{5}H^+ \rightarrow aCO_2 + bNH_4^+ + cH_3PO_4 + \frac{2a}{5}N_2 + \frac{7a}{5}H_2O$ | R2 |
| $OM^{\alpha,\beta} + 4aFe(OH)_3^\alpha + 4a\chi^\alpha Fe_{ox}P + 12aH^+ \rightarrow aCO_2 + bNH_4^+ + (c + 4a\chi^\alpha)H_3PO_4 + 4aFe^{2+} + 13aH_2O$ | R3 |
| $OM^{\alpha,\beta} + \frac{a}{2}SO_4^{2-} + aH^+ \rightarrow aCO_2 + bNH_4^+ + cH_3PO_4 + \frac{a}{2}H_2S + aH_2O$ | R4 |
| $OM^{\alpha,\beta} \rightarrow \frac{a}{2}CO_2 + bNH_4^+ + cH_3PO_4 + \frac{a}{2}CH_4$ | R5 |
| $CO_2 + 4H_2 \rightarrow CH_4 + 2H_2O$ | R6 |
| Secondary redox and other reaction equations† | |
| $2O_2 + NH_4^+ + 2HCO_3^- \rightarrow NO_3^- + 2CO_2 + 3H_2O$ | R7 |
| $O_2 + 4Fe^{2+} + 8HCO_3^- + 2H_2O + 4\chi^\alpha H_2PO_4^- \rightarrow 4Fe(OH)_3^\alpha + 4\chi^\alpha Fe_{ox}P + 8CO_2$ | R8 |
| $2O_2 + FeS \rightarrow SO_4^{2-} + Fe^{2+}$ | R9 |
| $7O_2 + 2FeS_2 + 2H_2O \rightarrow 4SO_4^{2-} + 2Fe^{2+} + 4H^+$ | R10 |
| $2O_2 + H_2S + 2HCO_3^- \rightarrow SO_4^{2-} + 2CO_2 + 2H_2O$ | R11 |
| $2O_2 + CH_4 \rightarrow CO_2 + 2H_2O$ | R12 |
| $2Fe(OH)_3^\alpha + 2\chi^\alpha Fe_{ox}P + H_2S + 4CO_2 \rightarrow 2Fe^{2+} + 2\chi^\alpha H_2PO_4^- + S_0 + 4HCO_3^- + 2H_2O$ | R13 |
| $2Fe(OH)_3^\beta + 2\chi^\beta Fe_{ox}P + H_2S + 4CO_2 \rightarrow 2Fe^{2+} + 2\chi^\beta H_2PO_4^- + S_0 + 4HCO_3^- + 2H_2O$ | R14 |
| $Fe^{2+} + H_2S \rightarrow FeS + 2H^+$ | R15 |
| $FeS + H_2S \rightarrow FeS_2 + H_2$ | R16 |
| $4S_0 + 4H_2O \rightarrow 3H_2S + SO_4^{2-} + 2H^+$ | R17 |
| $FeS + S_0 \rightarrow FeS_2$ | R18 |
| $SO_4^{2-} + CH_4 + CO_2 \rightarrow 2HCO_3^- + H_2S$ | R19 |
| $CH_4 + 8Fe(OH)_3^{\alpha,\beta} + 8\chi^{\alpha,\beta} Fe_{ox}P + 15H^+ \rightarrow HCO_3^- + 8Fe^{2+} + 8\chi^{\alpha,\beta} H_2PO_4^- + 21H_2O$ | R20 |
| $Fe(OH)_3^\alpha + (\chi^\alpha - \chi^\beta)Fe_{ox}P \rightarrow Fe(OH)_3^\beta + (\chi^\alpha - \chi^\beta)H_2PO_4^-$ | R21 |
| $Fe(OH)_3^\beta + (\chi^\beta - \chi^\gamma)Fe_{ox}P \rightarrow Fe(OH)_3^\gamma + (\chi^\beta - \chi^\gamma)H_2PO_4^-$ | R22 |
| $3Fe^{2+} + 2HPO_4^- \rightarrow Fe_3(PO_4)_2 + 2H^+$ | R23 |
| $Fe^{2+} + CO_3^{2-} \rightarrow FeCO_3$ | R24 |
| $FeCO_3 + H_2S \rightarrow FeS + HCO_3^- + H^+$ | R25 |
| $Fe_3(PO_4)_2 + 3H_2S \rightarrow 3FeS + 2HPO_4^{2-} + 4H^+$ | R26 |

* Organic matter (OM) is of the form $(CH_2O)_a(NH_4^+)_b(H_3PO_4)_c$, with 'a'=1, 'b' = 1/16 and 'c' = 1/106. Under anoxic bottom
water conditions, 'c' reduces to 0.25. † $\chi^{\alpha,\beta,\gamma}$ refers to the P:Fe ratio of $Fe(OH)_3^{\alpha,\beta,\gamma}$ (see Supplementary Table S1). $R6$ = $CO_2$
reduction; $R7$ = nitrification; $R8$ = Fe(OH)₃ formation; $R9$ = FeS oxidation; $R10$ = FeS₂ oxidation; $R11$ = H₂S oxidation; $R12$ =
aerobic CH₄ oxidation; $R13$ and $R14$ = Fe(OH)₃ reduction by H₂S; $R15$= FeS formation; $R16$ = pyrite formation (H₂S pathway);
$R17$ = S₀ disproportionation; $R18$ = pyrite formation (polysulfide pathway); $R19$ = SO₄-AOM; $R20$ = Fe-AOM; $R21$ = conversion
(i.e. crystallization) from α to β phase; $R22$ = crystallization from β to γ phase; $R23$ = vivianite formation; $R24$ = siderite
precipitation; $R25$ = conversion from siderite to FeS; $R26$ = vivianite dissolution by dissolved sulfide





**Table 4. Reaction equations implemented in the model.**

**Primary redox reaction equations**

$$R_1 = k_{\propto,\beta} OM^{\propto,\beta} \left( \frac{[O_2]}{K_{O_2}+[O_2]} \right) \tag{E1}$$

$$R_2 = k_{\propto,\beta} OM^{\propto,\beta} \left( \frac{[NO_3^-]}{K_{NO_3^-}+[NO_3^-]} \right) \left( \frac{K_{O_2}}{K_{O_2}+[O_2]} \right) \tag{E2}$$

$$R_3 = k_{\propto,\beta} OM^{\propto,\beta} \left( \frac{[Fe(OH)_3^\alpha]}{K_{Fe(OH)_3^\alpha}+[Fe(OH)_3^\alpha]} \right) \left( \frac{K_{NO_3^-}}{K_{NO_3^-}+[NO_3^-]} \right) \left( \frac{K_{O_2}}{K_{O_2}+[O_2]} \right) \tag{E3}$$

$$R_4 = \Psi k_{\propto,\beta} OM^{\propto,\beta} \left( \frac{[SO_4^{2-}]}{K_{SO_4^{2-}}+[SO_4^{2-}]} \right) \left( \frac{K_{Fe(OH)_3^\alpha}}{K_{Fe(OH)_3^\alpha}+[Fe(OH)_3^\alpha]} \right) \left( \frac{K_{NO_3^-}}{K_{NO_3^-}+[NO_3^-]} \right) \left( \frac{K_{O_2}}{K_{O_2}+[O_2]} \right) \tag{E4}$$

$$R_5 = \Psi k_{\propto,\beta} OM^{\propto,\beta} \left( \frac{K_{SO_4^{2-}}}{K_{SO_4^{2-}}+[SO_4^{2-}]} \right) \left( \frac{K_{Fe(OH)_3^\alpha}}{K_{Fe(OH)_3^\alpha}+[Fe(OH)_3^\alpha]} \right) \left( \frac{K_{NO_3^-}}{K_{NO_3^-}+[NO_3^-]} \right) \left( \frac{K_{O_2}}{K_{O_2}+[O_2]} \right) \tag{E5}$$

$$R_6 = k_1 DIC \left( \frac{K_{SO_4^{2-}}}{K_{SO_4^{2-}}+[SO_4^{2-}]} \right) \left( \frac{K_{Fe(OH)_3^\alpha}}{K_{Fe(OH)_3^\alpha}+[Fe(OH)_3^\alpha]} \right) \left( \frac{K_{NO_3^-}}{K_{NO_3^-}+[NO_3^-]} \right) \left( \frac{K_{O_2}}{K_{O_2}+[O_2]} \right) \tag{E6}$$

**Secondary redox and other reaction equations**

| | |
|---|---|
| $R_7 = k_2[O_2][NH_4^+]$ | (E7) |
| $R_8 = k_3[O_2][Fe^{2+}]$ | (E8) |
| $R_9 = k_4[O_2][FeS]$ | (E9) |
| $R_{10} = k_5[O_2][FeS_2]$ | (E10) |
| $R_{11} = k_6[O_2][\sum H_2S]$ | (E11) |
| $R_{12} = k_7[O_2][CH_4]$ | (E12) |
| $R_{13} = k_8[Fe(OH)_3^\propto][\sum H_2S]$ | (E13) |
| $R_{14} = k_9[Fe(OH)_3^\beta][\sum H_2S]$ | (E14) |
| $R_{15} = k_{10}[Fe^{2+}][\sum H_2S]$ | (E15) |
| $R_{16} = k_{11}[FeS][\sum H_2S]$ | (E16) |
| $R_{17} = k_{12}[S_0]$ | (E17) |
| $R_{18} = k_{13}[FeS][S_0]$ | (E18) |
| $R_{19} = k_{14}[SO_4^{2-}][CH_4]$ | (E19) |
| $R_{20} = k_{15}[Fe(OH)_3^{\propto,\beta}][CH_4]$ | (E20) |
| $R_{21} = k_{16}[Fe(OH)_3^\propto]$ | (E21) |
| $R_{22} = k_{17}[Fe(OH)_3^\beta]$ | (E22) |
| $R_{23} = k_{18}[Fe^{2+}][HPO_4^{2-}]$ | (E23) |
| $R_{24} = k_{19}[Fe^{2+}][DIC]$ | (E24) |
| $R_{25} = k_{20}[FeCO_3][\sum H_2S]$ | (E25) |
| $R_{26} = k_{21}[Fe_3(PO_4)_2][\sum H_2S]$ | (E26) |





**Table 5. Reaction parameters used in the diagenetic model.**

| Parameter | Symbol | Value | Units | Values given in literature |
|---|---|---|---|---|
| Decay constant for $C_{org}^{\alpha}$ | $k_{\alpha}$ | 0.05 | yr$^{-1}$ | 0.05-1.62[a,b] |
| Decay constant for $C_{org}^{\beta}$ | $k_{\beta}$ | 0.0086 | yr$^{-1}$ | 0.0086[b] |
| Limiting concentration of $O_2$ | $K_{O2}$ | 0.02 | mM | 0.001-0.03[c] |
| Limiting concentration of $NO_3^-$ | $K_{NO3^-}$ | 0.004 | mM | 0.004-0.08[c] |
| Limiting concentration of $Fe(OH)_3$ | $K_{Fe(OH)3}$ | 65 | µmol g$^{-1}$ | 65-100[c] |
| Limiting concentration of $SO_4^{2-}$ | $K_{SO42-}$ | 1.6 | mM | 1.6[c] |
| Attenuation factor for $SO_4^{2-}$ and methanogenesis | $\Psi$ | 0.0042 | - | 0.00157-0.075[b,d] |
| Rate constant for reaction *E6* | $k_1$ | 0.0011 | yr$^{-1}$ | |
| Rate constant for reaction *E7* | $k_2$ | 10'000 | mM$^{-1}$ yr$^{-1}$ | 5'000-39'000[c,d] |
| Rate constant for reaction *E8* | $k_3$ | 140'000 | mM$^{-1}$ yr$^{-1}$ | 140'000[c] |
| Rate constant for reaction *E9* | $k_4$ | 300 | mM$^{-1}$ yr$^{-1}$ | 300[c] |
| Rate constant for reaction *E10* | $k_5$ | 1 | mM$^{-1}$ yr$^{-1}$ | 1[c] |
| Rate constant for reaction *E11* | $k_6$ | 160 | mM$^{-1}$ yr$^{-1}$ | $\geq$ 160[c] |
| Rate constant for reaction *E12* | $k_7$ | 10'000'000 | mM$^{-1}$ yr$^{-1}$ | 10'000'000[c] |
| Rate constant for reaction *E13* | $k_8$ | 9.5 | mM$^{-1}$ yr$^{-1}$ | $\leq$ 100[c] |
| Rate constant for reaction *E14* | $k_9$ | 0.95 | mM$^{-1}$ yr$^{-1}$ | Model constrained |
| Rate constant for reaction *E15* | $k_{10}$ | 150 | mM$^{-1}$ yr$^{-1}$ | 100-14'800[b, d] |
| Rate constant for reaction *E16* | $k_{11}$ | 0.0003 | mM$^{-1}$ yr$^{-1}$ | 3.15[e] |
| Rate constant for reaction *E17* | $k_{12}$ | 3 | yr$^{-1}$ | 3[f] |
| Rate constant for reaction *E18* | $k_{13}$ | 1 | mM$^{-1}$ yr$^{-1}$ | 7[f] |
| Rate constant for reaction *E19* | $k_{14}$ | 0.14 | mM$^{-1}$ yr$^{-1}$ | 10[c] |
| Rate constant for reaction *E20* | $k_{15}$ | 0.00000016 | mM$^{-1}$ yr$^{-1}$ | 0.0074[g] |
| Rate constant for reaction *E21* | $k_{16}$ | 0.6 | yr$^{-1}$ | 0.6[f] |
| Rate constant for reaction *E22* | $k_{17}$ | 0.000013 | yr$^{-1}$ | Model constrained |
| Rate constant for reaction *E23* | $k_{18}$ | 0.052 | mM$^{-1}$ yr$^{-1}$ | Model constrained |
| Rate constant for reaction *E24* | $k_{19}$ | 0.0027 | mM$^{-1}$ yr$^{-1}$ | Model constrained |
| Rate constant for reaction *E25* | $k_{20}$ | 0.0008 | mM$^{-1}$ yr$^{-1}$ | Model constrained |
| Rate constant for reaction *E26* | $k_{21}$ | 0.0008 | mM$^{-1}$ yr$^{-1}$ | Model constrained |

[a] Moodley et al. (2005); [b] Reed et al. (2011a); [c] Wang and Van Cappellen (1996); [d] Reed et al. (2011b); [e] Rickard and Luther
(1997); [f] Berg et al. (2003); [g] Rooze et al. (2016)





**Table 6. Depth-integrated rates of key processes for selected depth intervals in µmol m$^{-2}$ d$^{-1}$.**

| Process | 0 – 90 cm[a] | 90 - 300 cm[b] | 300 – 800 cm[c] | 0 – 800 cm |
|---|---|---|---|---|
| Organoclastic $SO_4^{2-}$ reduction | 698.12 | 22.20 | 0.012 | 720.34 |
| Methanogenesis (OM) | 18.81 | 12.02 | 46.24 | 77.07 |
| Methanogenesis (DIC) | 0.35 | 17.24 | 40.33 | 57.92 |
| $SO_4$ - AOM | 10.05 | 157.42 | 1.37 | 168.83 |
| Fe – AOM[d] | 0 | 0 | 0.14 | 0.14 |
| $S_0$ disproportionation | 0 | 0 | 1.13 | 1.13 |

[a] Marine deposits ; [b] limnic sediments around the SMTZ with dissolved sulfide; [c] non-sulfidic limnic deposits; [d] per mol of $CH_4$





**Figures**

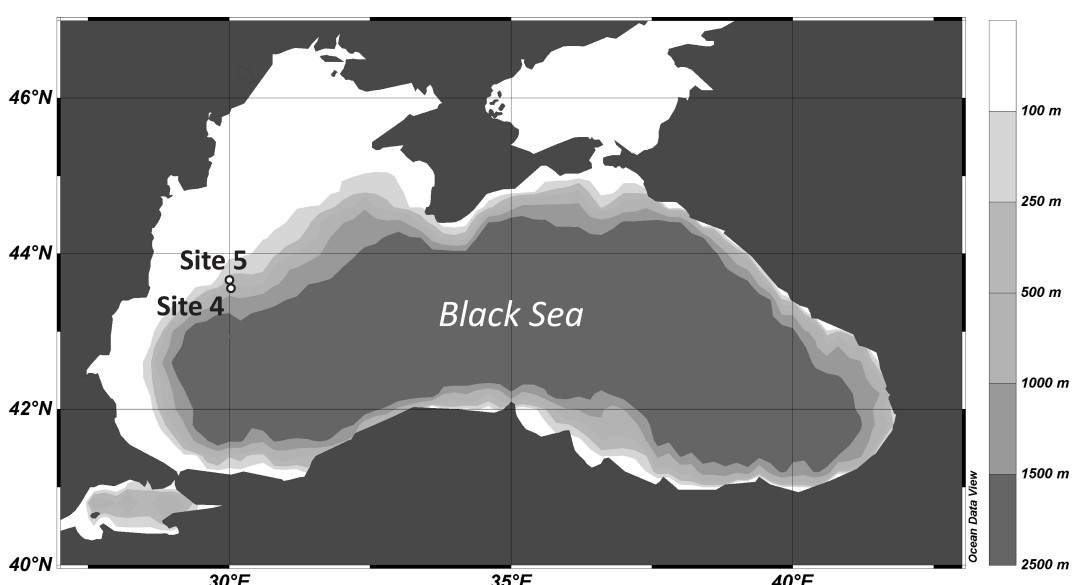


**Figure 1. Map showing the locations of site 4 (43°40.6' N, 30°7.5' E; 377 mbss) and site 5 (43°42.6' N, 30°6.1' E; 178 mbss),**
**sampled in June 2013.**






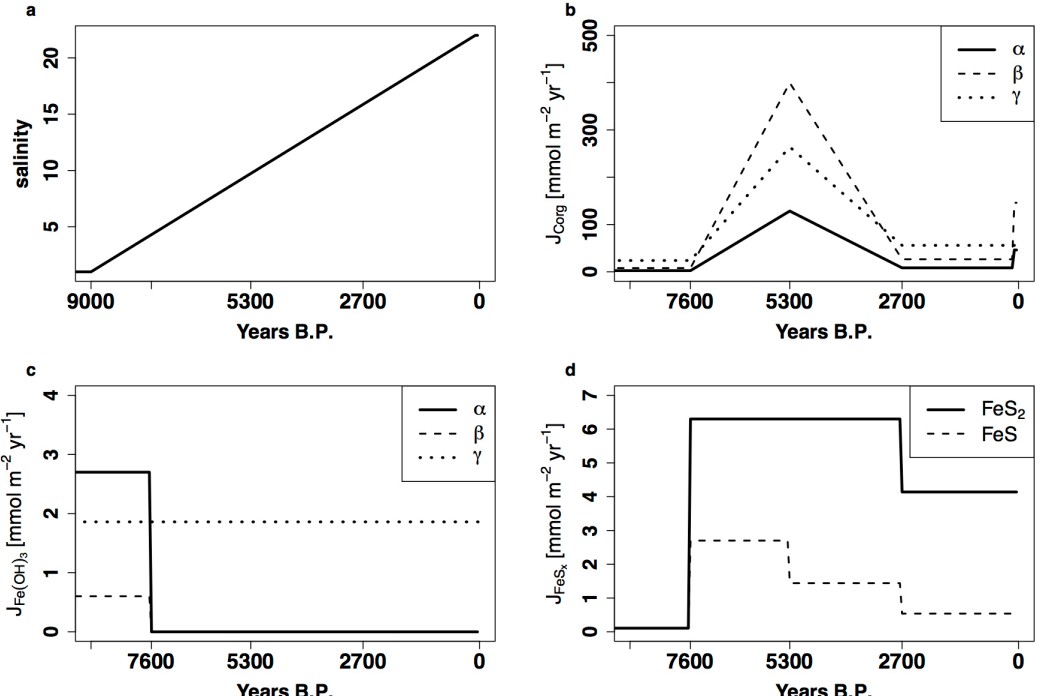


**Figure 2. Transient evolution of salinity with a linear increase from 1 to 22 between 9000 and 100 years B.P. (a), fluxes of**
**organic matter ($J_{C_{Org}}$; b), Fe oxides ($J_{Fe(OH)_3}$; c) and Fe sulfides ($J_{FeS_x}$; d) as implemented in the diagenetic model (site 4).**





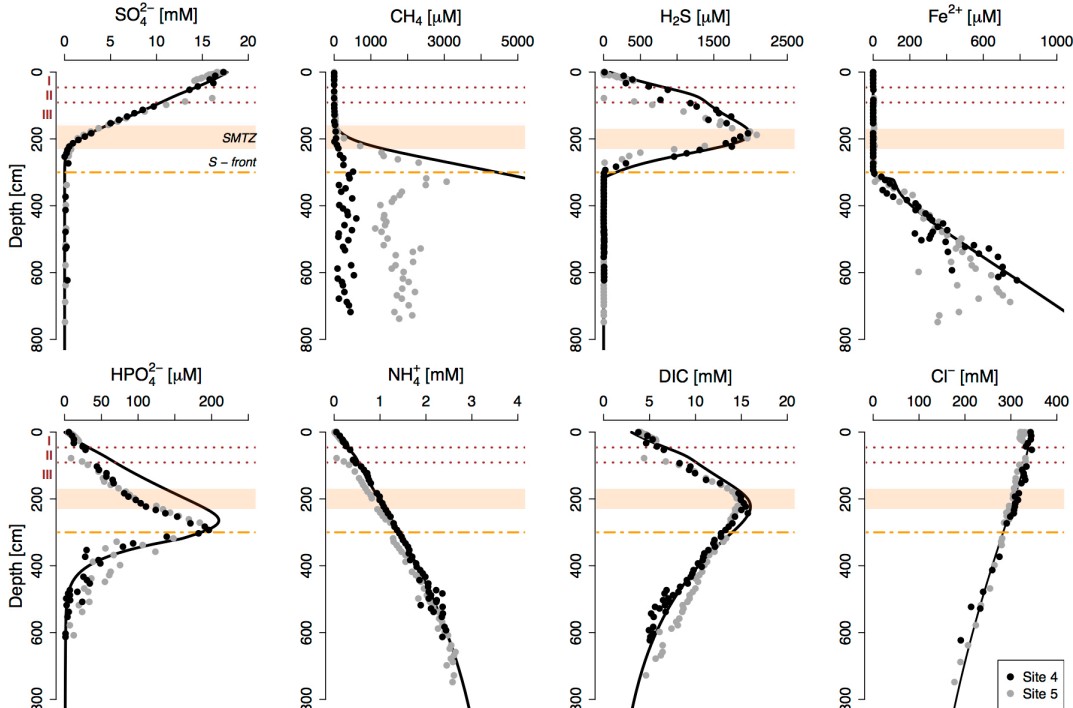


**Figure 3. Pore water profiles of key components for site 4 (black dots) and site 5 (gray dots) and corresponding modeled profiles as calculated with the diagenetic model (black lines). Red dotted lines and roman numbers indicate the transitions between the lithological Unit I (modern coccolith ooze), Unit II (marine sapropel) and Unit III (limnic deposits). The orange bar represents the sulfate-methane transition zone (SMTZ) and the orange dashed line shows the current position of the downward migrating sulfidization front (S-front).**

867

868



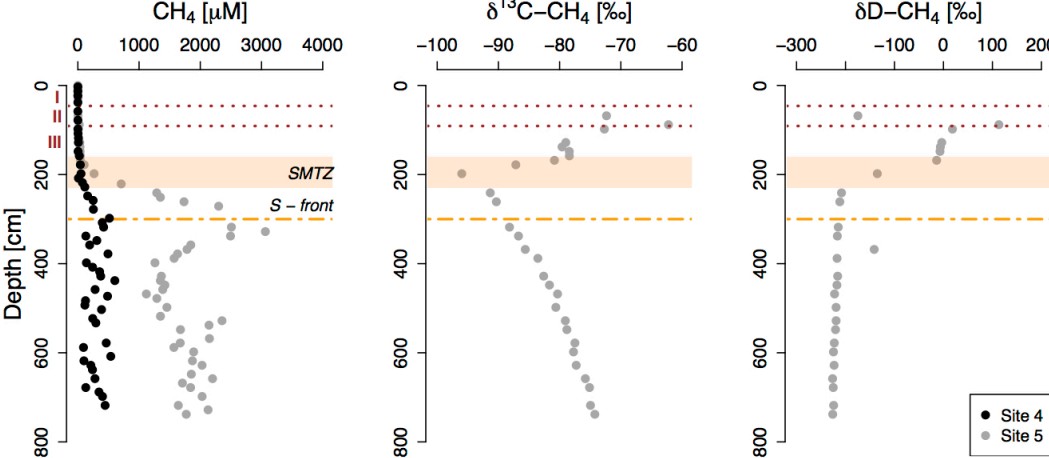

**Figure 4.** Pore water profiles of CH$_4$ for site 4 (black dots) and 5 (gray dots) and corresponding isotopic composition of dissolved CH$_4$ (available for site 5 only). δ$^{13}$C-CH$_4$ values are given in ‰ vs. VPDB (Vienna Pee Dee Belemnite) and δD-CH$_4$ values are given in ‰ vs. V-SMOW (Vienna Standard Mean Ocean Water). Red dotted lines and roman numbers indicate the transitions between the lithological Unit I (modern coccolith ooze), Unit II (marine sapropel) and Unit III (limnic deposits). The orange bar represents the sulfate-methane transition zone (SMTZ) and the orange dashed line shows the current position of the downward migrating sulfidization front (S-front).





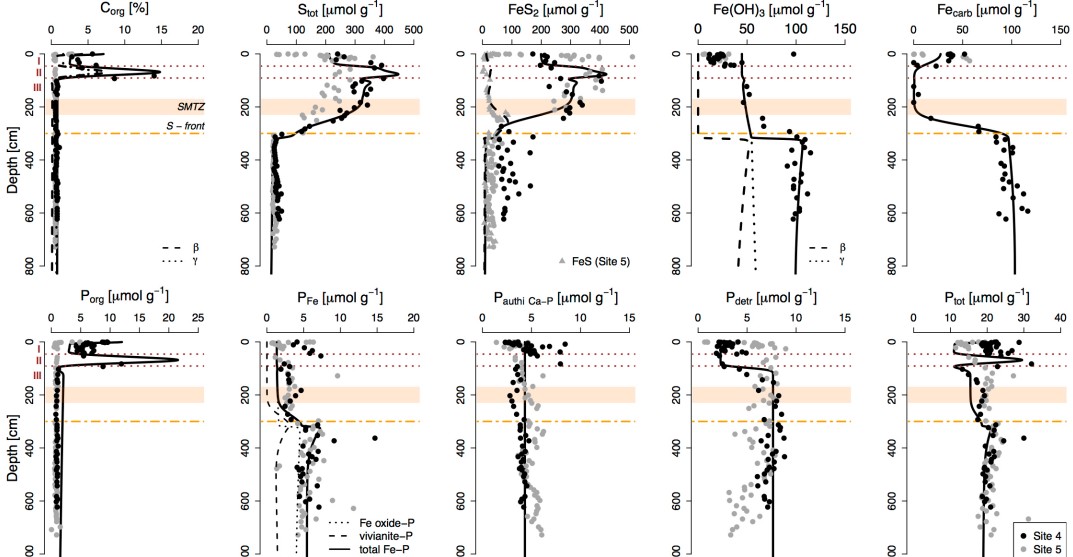


**Figure 5. Solid phase sediment profiles for site 4 (black dots) and 5 (gray dots). Fe oxides represent the sum of amorphous,**
**crystalline and recalcitrant oxides, i.e. $Fe_{ox1}$, $Fe_{Ox2}$ and $Fe_{mag}$ (Table 1, Supplementary Fig. S2). $Fe_{carb}$ was corrected for**
**apparent AVS dissolution during the Na acetate extraction step (the uncorrected $Fe_{carb}$ data is given in Supplementary**
**Fig. S2). Black lines represent profiles derived from the diagenetic model. Red dotted lines and roman numbers indicate**
**the transitions between the lithological Unit I (modern coccolith ooze), Unit II (marine sapropel) and Unit III (limnic**
**deposits). The orange bar represents the sulfate-methane transition zone (SMTZ) and the orange dashed line shows the**
**current position of the downward migrating sulfidization front (S-front).**





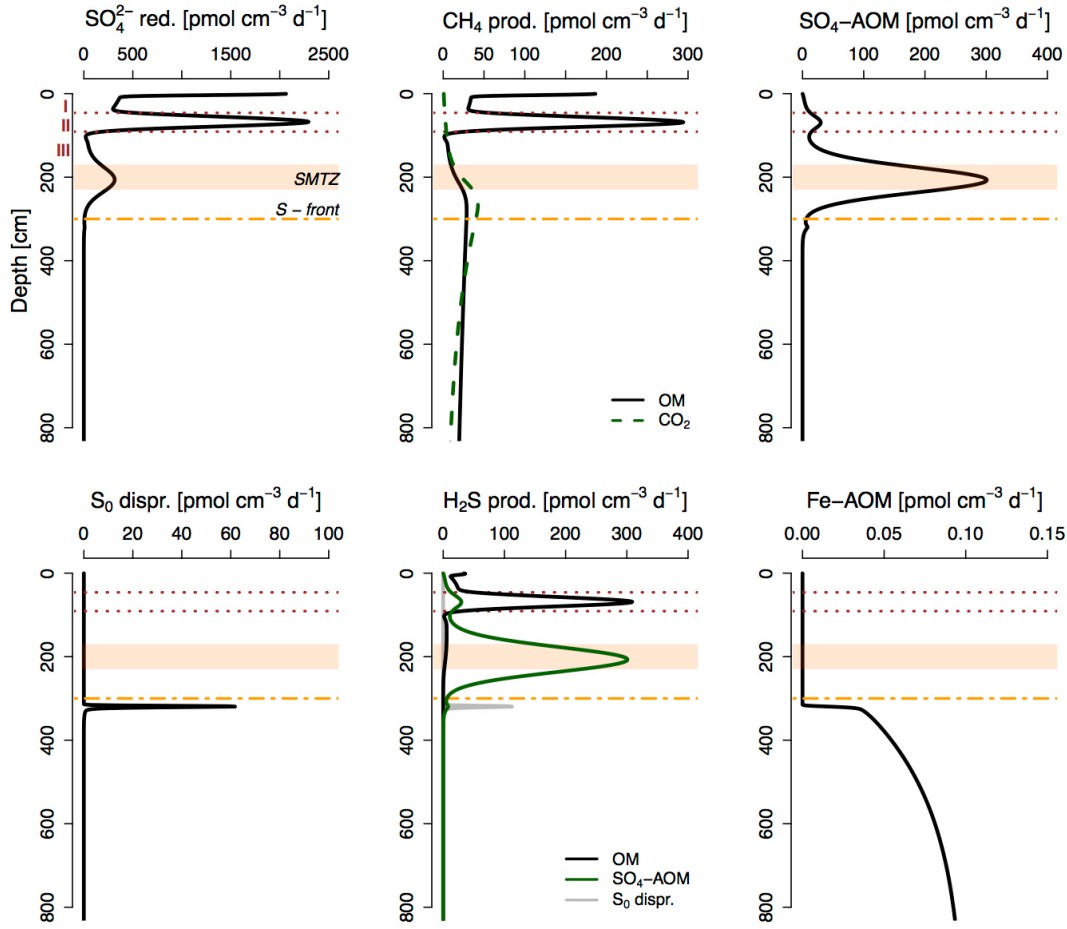

**Figure 6. Modeled rates of total SO$_4^{2-}$ reduction, methanogenesis, SO$_4$-AOM, S$_0$ disproportionation, sulfide production and Fe-AOM. Methanogenesis is divided into CH$_4$ production from organic matter fermentation ("OM"; black solid line) and CO$_2$ reduction ("CO$_2$"; green dashed line). Red dotted lines and roman numbers indicate the transitions between the lithological Unit I (modern coccolith ooze), Unit II (marine sapropel) and Unit III (limnic deposits). The orange bar represents the sulfate-methane transition zone (SMTZ) and the orange dashed line shows the current position of the downward migrating sulfidization front (S-front).**





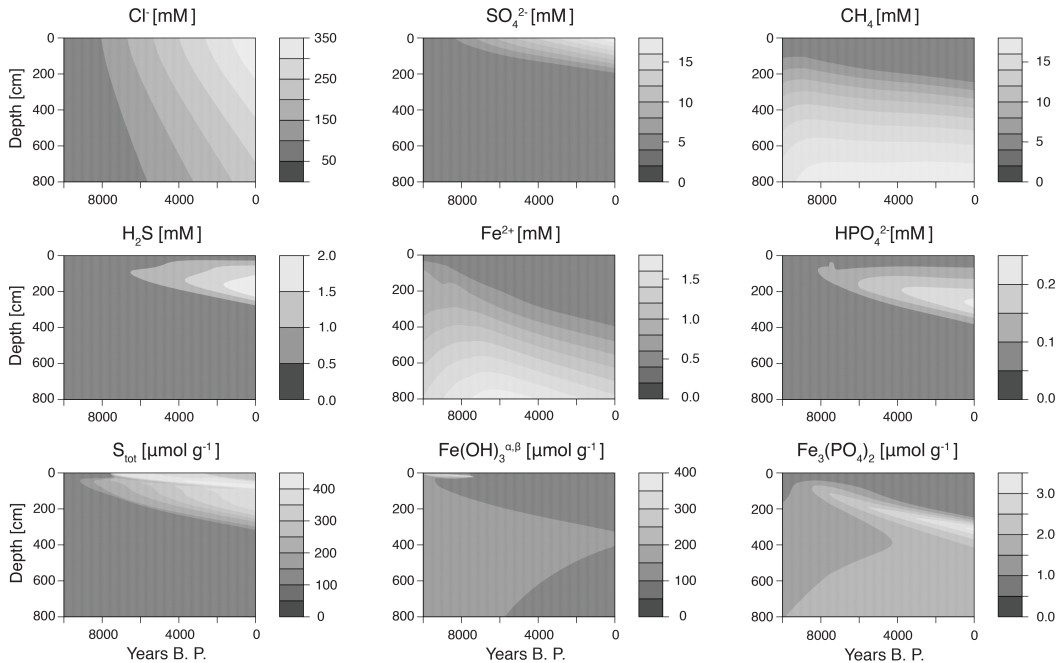


**Figure 7. Transient evolution of selected pore water and sediment profiles with depth as calculated for site 4 using the**

**diagenetic model.**







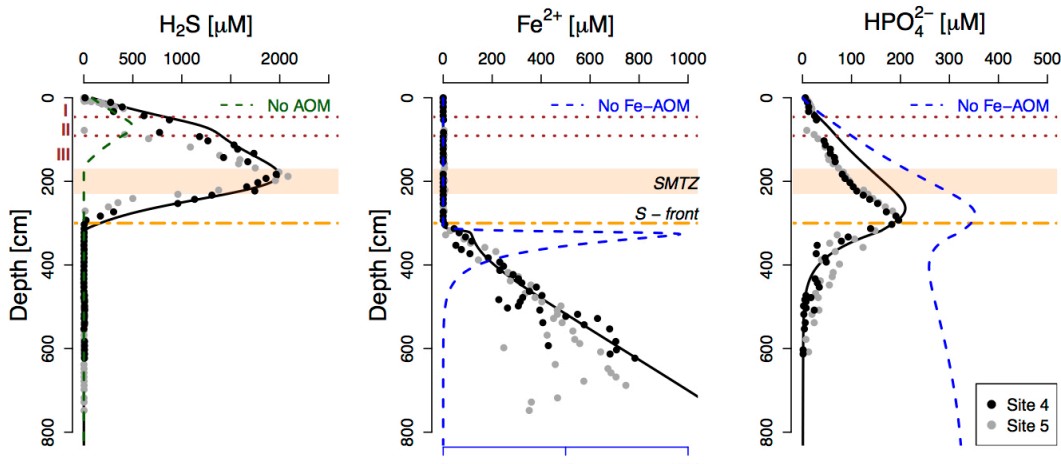


**Figure 8. Pore water profiles of dissolved sulfide, $Fe^{2+}$ and $HPO_4^{2-}$. The green dashed line represents the modeled sulfide**
**profile without SO4-AOM, indicating that latter significantly enhances the downward sulfidization. Blue dashed lines**
**denote the modeled Fe2+ and HPO42- profiles without ongoing Fe oxide reduction in the limnic deposits (i.e. no Fe-AOM).**
**Note that concentrations of Fe2+ were multiplied 10 times in the model simulation without Fe oxide reduction to better**
**visualize the potential release of Fe2+ through a cryptic S cycle (corresponding x axis at bottom). Red dotted lines and**
**roman numbers indicate the transitions between the lithological Unit I (modern coccolith ooze), Unit II (marine sapropel)**
**and Unit III (limnic deposits). The orange bar represents the sulfate-methane transition zone (SMTZ) and the orange**
**dashed line shows the current position of the downward migrating sulfidization front (S-front).**



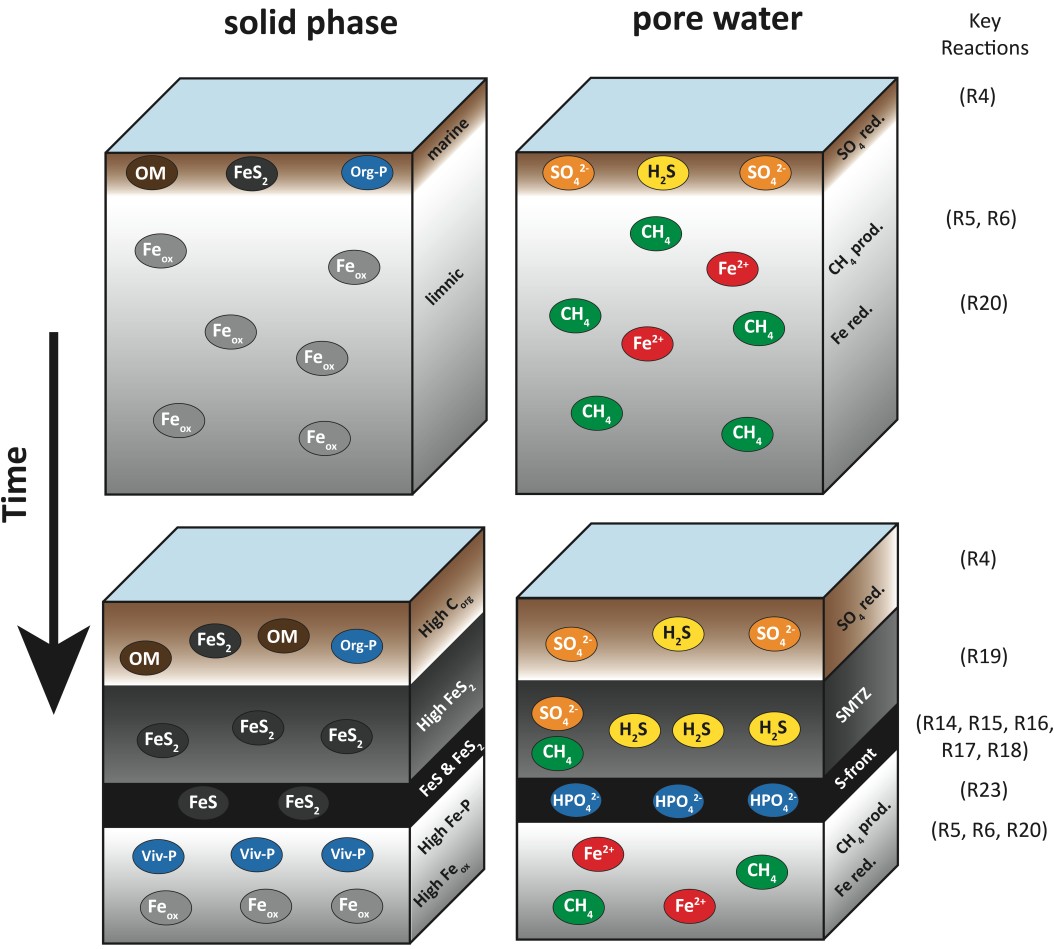


**Figure 9. Schematic of the main diagenetic processes discussed in this study and their imprint on the geochemical solid phase (left) and pore water profiles (right). Accumulation of marine sediments with time and the subsequent downward diffusion of $SO_4^{2-}$ into the $CH_4$-bearing limnic sediment stimulate $SO_4$-AOM around the sulfate-methane transition zone (SMTZ), thus enhancing the downward sulfidization of the Fe oxide-rich lake deposits. Below the sulfidization front (S-front), $HPO_4^{2-}$ released during reductive dissolution of Fe oxides is bound again in vivianite, leading to an enrichment in sedimentary P in these sediments. Numbers on the right indicate the key reactions occurring in the corresponding sediment layers as described in Table 3. Note that in this study, Fe-AOM (R20) represents the main source of pore water $Fe^{2+}$ below the S-front. However, based on the geochemical data, we cannot exclude a potential role for organoclastic Fe reduction (R3).**