# Peer review of "Anaerobic oxidation of methane alters sediment records of 1 sulfur, iron and phosphorus in the Black Sea 2"

_Biogeosciences, 2016_

## Referee Comment (RC1) · O. Sivan (Referee) · 24 Mar 2016

O. Sivan (Referee)

oritsivan@gmail.com

Please see the comments also in the attached file (as supp)

General comments This paper deals with diagenetic processes in the sediments of the Black Sea which changed from a lacustrine environment to a marine system. The work focuses on AOM and its effect on the linked species and processes under these changes. This was done by producing solid phase and porewater profiles, and by diagenetic modeling. The work is well written and easy to read, and I found it complete, serious and convincing. The authors measured, calculated and thought on almost all the possible aspects that could affect this system during these changes. This careful work enhances our understanding on AOM by iron and sulfate in marine setting in

general and specifically in a complex setting. It also provides us with new knowledge on the Black Sea's limnological history. I thus suggest accepting this work pending minor comments. The main comments to the authors: • The model is not detailed and explained enough. You should cite less Rooze et al 2016 and provide more details here. Also, you should perform sensitivity tests for the various uncertainties. I did not have access to Rooze et al paper, but from its title I am assuming it is not on the same system so there is no overlapping. You should however upload this paper. • The Fe2+ increase in the deep sediments could be from deep Fe-AOM as we see in lakes and coastal sediments (Sivan et al., 2011; Egger et al., 2015), however it can also be organoclastic. There may be reactivation of less soluble Fe(III) minerals in this system by other means other than methane oxidation (e.g as described by Sivan et al., 2016). You indeed mention it, however, you should refer to it as a possible option. • The assumption that the total dissolved Fe and Mn (as measured by AE-ICP) are Fe(II) and Mn(II) is probably fine, however you should test it and show it in at least in one of the profiles in the Black sea sediments (or cite other works that did it there). You should compare the Fe(II) to Fe(total) by another method (as the Ferrozine), or compare your assumed Fe(II) to Fe(II) from the Ferrozine or another method. • You should discuss in more detail the sulfate profile and – its apparent "diffusion" profile (linear curve) with organoclastic sulfate reduction, and the cryptic S cycle in the upper part of this profile. You should also compare the downward flux of sulfate to the SMTZ and the upward flux that you calculated for methane and discuss it. • The $\delta$13C of methane similarity to Yoshinaga's data is convincing and satisfactory. Interpretation/speculations regarding the profile of $\delta$D of methane should be given.

Specific comments • L 84-85: Vivanite was found also in Lake Kinneret (Sivan et al., 2011), it can support your finding and related processes. • L 92-93: Also propose the other option for Fe reduction. • L 141-142: See comment regarding this method above. • L. 151: I assume the auto analyzer was based on IR. How did you remove of the sulfide? • L. 236-242: Clarify and explain this part in more details. • L. 288: Show how you calculated to this saturation value and under which salinity

conditions. Mark this value on the figures of methane too. • Add the bottom water values on the porewater profiles • L 297: Change the sentence to a more precise one. • L. 381: Explain the other 91% based on the profile (see main comment). • L 445-451: See the main comment regarding organoclastic Fe reduction. • L. 566-568: You don't need this trivial sentence, your work is good and nice enough without it . • Fig. 3: No sulfate measurements in the sapropel depths ? Add saturation of methane. What could be the reaction precipitating phosphate in the upper 300 cm (hydroxyl-apatite)? • Fig. 4: Again with the saturation of methane. Technical comments • L 339: Start a new subchapter.

Please also note the supplement to this comment:
http://www.biogeosciences-discuss.net/bg-2016-64/bg-2016-64-RC1-supplement.pdf

---

## Referee Comment (RC2) · Anonymous Referee #2 · 6 Apr 2016

This study addresses the diagenetic implications of anaerobic methane oxidation in Black Sea sediments where marine deposits overlie a freshwater facies into which a sulfate front is advancing. High-resolution geochemical profiles of dissolved and solid species are presented from two adjacent sites, and the profiles are simulated in a complex non-steady-state diagenetic model that derives rates of the relevant processes.

The subject is interesting, obviously relevant to Biogeosciences, and the results and conclusions presented here are novel and add substantially to our understanding of sediment biogeochemistry and diagenesis. The text is well written, clear, and concise, the data is of good quality, and the conclusions are generally justified by the data and modelling. The authors particularly deserve credit for clearly distinguishing model

results from reality.

My main concern with the paper is that I miss a deeper analysis and discussion of the extent to which the modelling results are forced by the formulation and parameterization of the model. This could involve a sensitivity analysis or testing of alternative scenarios. Additionally, some aspects of the model results and formulation require clarification.

It is particularly the conclusions concerning sulfate- and iron-coupled AOM that require attention. The occurrence of Fe-AOM appears to be forced by the exclusion of organoclastic Fe reduction from the model, although there is plenty of evidence that organotrophic microbes can reduce crystalline Fe oxides, and there is no evidence that organotrophic Fe reduction cannot co-occur with methanogenesis if Fe reduction is limited by the availability/reactivity of iron oxides. Furthermore, is seems that partitioning of AOM must be sensitive to the parameterization of the pathways, which therefore needs to be discussed.

Specific issues: 22-23+89-90: The finding that sulfate-AOM enhances the sulfide flux is not novel according to lines 72-75.

289-96: Just a comment: The difference in the two methane profiles is strange and it is difficult to understand how degassing would have caused a proportional decrease in methane in the zone above the zone of saturation. Nonetheless, I agree that it is the most likely explanation given the similarity of all other profiles, including the methane isotopes. I suggest rephrasing 293-294 to clarify which methane data this applies to.

339-41+Fig 6: I don't understand the very high rates of sulfate reduction and methanogenesis in the sapropel, and the model doesn't seem to fit the data well here. Albeit noisy, the measured H2S profile seems straight or even concave in this region, and the same clearly goes for DIC, whereas the model profiles are convex, which suggests that the model overestimates the rates substantially. Although this zone is not of primary interest in this study, an overestimation of rates and product concentrations results in a shallower gradient from unit II to the SMTZ and therefore in lower sulfate-AOM rates,

so the fit here still influences the central conclusions.

Fig. 6, further+ Fig. 3: There seems to be an error in the H2S production panel in Fig. 6 as H2S production from sulfate reduction is only a fraction of the sulfate reduction rate? Shouldn't hese be 1:1 as is the case for sulfate-AOM and sulfide production from sulfate AOM? Also, what happens to methane produced in unit II? The methane profile appears flat, yet only a fraction of the production is consumed by AOM. Please provide blow-ups of modelled methane in the upper 2 m and of sulfate below the SMTZ in Fig. 3.

391-5: This is the only real flaw in the paper. The Rayleigh function applies to closed systems and should never be used in open systems such as this one, where diffusion affects the relative distribution of the isotopes. Accurate enrichment factors can only be derived through modelling (e.g., Alperin et al. 1988). The closed-system approach will underestimate enrichment factors substantially in most cases, and likely explains the low value derived here. This problem was described decades ago (e.g., Jørgensen 1979, GCA 43:363).

401: I think some of these studies observed sulfate reduction and did not only postulate it?

442-3: Under which conditions, if any, within a realistic parameter space or with an alternative set of reactions, would a cryptic sulfur cycle be able to explain the accumulation of Fe2+?

450-5: The references listed here suggest that AOM may be coupled to Fe reduction, but here you really use them to support the assumption that Fe reduction can be coupled to AOM rather than to organoclastic Fe reduction – Is there any support for that in any of those references? As stated in l. 463, organoclastic Fe reduction is clearly limited at these depths, but that doesn't mean that it is absent. Furthermore l. 474-6 seems to suggest organoclastic Fe reduction anyway, even if it is by archaea? But what special skills do these organisms have that would enable them to reactivate Fe oxides?

489-93: It seems trivial that in situ rates under the given conditions are low compared to lab-based rates. What is the observational basis for the parameterization of the reaction?

494-5: How sensitive is the sulfate/Fe-AOM partitioning to the parameterization?

Table 3, R6+16: I understand that you need a sink for H2, but why is it only methanogenesis and not, at least sulfate reduction? This will lead to overestimation of methanogenesis in the sulfate zone.

Table 4: R19+ R20 are biological processes and as such might obey biological (saturation) kinetics? These are key reactions in the paper and the observational basis for the kinetic expressions, and their impact on the conclusions should be discussed. Table 6 + Fig 6: The labelling of the two kinds of methanogenesis is misleading. The light isotopic composition of methane implies that it is formed mainly through CO2 reduction rather than acetoclastic methanogenesis, i.e. that "Methanogenesis (OM)" is mainly CO2-based. "Methanogenesis (DIC)" is really a peculiarity of the model and completely and uniquely linked to pyrite formation, so "Methanogenesis (FeS2)" would be more appropriate (but see also comment to Table 3 above).

Fig. 7: Consider a colour version here. The darkest shading on the scale bars always appears darker than the darkest part of the figures. Because the shading varies so little from min. to max. it is very difficult to extract quantitative information.

---

## Referee Comment (RC3) · W.-L. Hong (Referee) · 15 Apr 2016

Overall comments In the paper, Egger et al. present very comprehensive porewater and sediment geochemical data to discuss the cycles of C, S, Fe, and P in sediments that are unique in the sense of their depositional sequence and history. The authors collectively discuss the data with a rather complicated model, which is understandable due to the complexity of the system they work with. The paper is well-written and structured. The authors' attempt to elucidate the mass balance of several elements in this environment provide valuable insights to the coupling of these elements and the complexity of it. In general, the conclusions are convincing and well support by their data and modeling. As also a user of such transport-reaction models (Hong et al.,

2014a&b; Torres et al., 2015; Hong et al., 2016), my major concern of the paper will be on the assumptions and setup of the model as well as several interpretations the authors made based on the model results. In conclusion, I think this is a very nice piece of work considering all the data and modeling work.

Major comments Modeling numerical issue: In conventional models for transport-reaction models, advection (i.e. sediment burial) often inevitably results in numerical dispersion, concentration will decrease as time progresses with burial even without any reaction. This effect will be especially obvious when using high advection rate (burial rate), large time discretization, and a long modeling time. I've done some tests before (Hong et al., 2016 accepted by limnology and oceanography) by simulating time progression of a profile with sharp concentration change. After 140 years of simulation, the concentration is 20% reduced compared to the value it should have (see the attached file for this comparison). As for the sharp increase of OM content in your environment, you will inevitably encounter this numerical issue. I urge the authors to run some simulations with only burial (no diffusion and other reactions), and see how your sediment and porewater profiles will progress.

I also wonder what is the consequence to accelerate your model. The price of numerical issues will be greater when you accelerate it by using larger time steps and/or faster rates. It is almost no way to have a model that is both efficient and accurate. There is always a sacrifice.

The other potential numerical issue I want to point out for the authors is the convergence of the model results. You have to make sure you use temporal and spatial steps small enough so that the results are stable. This can be done by running the model several times (the same reactions and setup) with smaller time/space discretization for each run. Chose the smallest discretization that your model results stop changing.

Conflict between observations and model results: In a few places in the paper, the authors didn't explain clearly the conflict between observations and model results.

One example is the choice of chloride changes with time. The authors used a very different evolution pattern from what literature suggested because it provides a better fit of their chloride concentration. However, the time scale adopted by the authors (100yrs) is an order smaller than what is suggested in the literature (2000yrs). The authors provide no explanation about such difference. I envision that if the author use constant chloride from 2000yrs BP until now and increase fluid advection rate (larger u), they might be able to fit the profile. I think the authors should explain better why choosing such condition.

The other example is from line 369 to 371. The authors claimed the SR rates they estimated from the model in zone I & II are more correct the estimation from porewater profiles. This statement raises the question that, then how do you know the SR rate you estimated from these two zones are accurate since you have no data to support you.

Very high methanogenesis rate in sulfate reduction zone: In fig 6, there are two peaks of methane production (one in bottom water or first cm of sediment? While the other in zone II). My questions are two: 1) It is apparent that this methanogenesis is from OM decomposition. However, methanegenesis should be suppressed when the sulfate content is high, as in the case of zone II. I understand that although methanogenesis is inhibited by sulfate content (E5 & E6 in Table 4), model can still produce very high ME rate when there is ultrahigh OM content. However, a model is a model, do you have any prove such high methanegenesis from your zone I and II. Considering the CH4 production rate and SO4-AOM rate from Fig.6. you should see either high methane or light d13C of methane in zone I and II if the rate is this high. I however don't quite see those from your profiles.

2) Back to fig6, your rates do not seem to balance. The highest CH4 production rate approaches 300 pmol/cm3/d which only stimulates an AOM rate less than maybe 20 pmol/cm3/d. If there is more production than consumption, isn't that you will methane accumulates in the porewater (i.e., high methane from that depth in the sediments).

SR rate is over 2000 pmol/cm3/d in this section but sulfide production is only 300 pmol/cm3/d. where is the rest of sulfide production?

The very complicated model: The authors use a rather complicated model in this study by choosing many reactions that are not totally necessary. For example, the authors choose to include aerobic processes (R1, R7-R12) and nitrate reduction (R2) even though there is no constraints on O2 and nitrate content in the porewater. I would also doubt the importance of these reactions due to the anoxic bottom water in Black Sea. The authors chose not to include Mn reduction due to its low content, which is fine with me, but decide to include all other processes that cannot be constrained? That is an odd decision to me. By excluding these unnecessary reactions, the authors can also improve the efficiency of the model.

I also wonder, with all the reactions assigned in the model, do the authors have enough constraints? I believe the answer should be close to yes as the authors have many data to support the model (which is very nice). I would urge the authors to spare a section in the text discussing the constraints for the model. To me, this is an extremely important but often ignored aspect in papers like this. I have done some initial analyses based on the reaction network in Table 3. For example, for Fe2+, the authors have R3, R9, R10, R13, R14, and R20 for sources, and R8, R15, R23, and R24 for sinks. Some source and sink terms may be constrained by the data of iron mineral speciation. When the same analyses being applied on HS in porewater, it seems like the abundance of different Fe-S minerals also depend on the source and sink terms of HS. A table such as tab6 but with more species included may be useful for such discussion.

One last comment on the complicated model, how does the model describe pH, which should be very important determining the type of dissolved sulfide and DIC. I don't see reactions such as H2S becomes HS-+H+ in Tab3 which describe the buffer capability of HS species (need same reactions for carbonate systems) . Although there is usually no good constraint on pH, it's good to make sure pH falls in the right range especially when including pH-sensitive reactions.

Minor comments Line 151: Please specify how you measure sulfide, phosphate, and DIC onboard. General model: What is the initial condition? Line 211: why 20 meters? You should mark you explain this in the supplemental material Line 255: is zero gradient a good assumption for methane? How do you know there is no deeper source of methane Line 289 to 293: You have same ammonium but higher methane in site4 and 5. Of course more severe degassing during core recovery in site 4 can be one explanation, but maybe there is more methane input from site 5 from greater depth. This echoes back my previous comment: is zero gradient really a good assumption for methane? Line 304: How do you know the isotopic signature of methane is not affected by degassing? Line 533: Isn't that this will be capture in your orgP analyses?

Line 827: Do you have any constrain on C/P ratio? Maybe this explains why the fitting on porewater phosphate profile is not as good?

Salinity/chloride: In many places of the paper, the authors mixed the term salinity and chloride concentration (e.g., line268-282). Of course these two properties are usually linear dependent on each other but they are fundamentally different and may correlate with each other very differently when Black Sea was more of a "lake" or a "Sea". I suggest the authors to use chloride concentration throughout the paper or explain how they convert salinity to chloride concentration.

FigS3: What is going on with the very high alkalinity at very top? where dic concentration looks normal...

Please also note the supplement to this comment:
http://www.biogeosciences-discuss.net/bg-2016-64/bg-2016-64-RC3-supplement.pdf
* * *
[Figure]

**Supplement:**

*CrunchFlow numerical issue in advection*

We did not use the CrunchFlow built-in function to describe fluid advection due to a significant numerical dispersion in CrunchFlow. This numerical issue is confirmed both by CrunchFlow developer (Steefel, personal communication) and also the tests we performed. Here we compared two simulations done by the CrunchFlow built-in function for advection (the "Erosion/burial" function) and our MATLAB routine to demonstrate this numerical issue. Both simulations were done assuming a 4-meter sediment column. Constant porosity of 0.68 throughout the core was used. We turned off all reactions and diffusion; only advection of fluid was allowed. Fixed concentrations for the upper boundary condition was used; a no flux lower boundary condition was adopted. Three lengths of simulation time were chose: 20, 80 and 140 years. The maximum allowed time step in CrunchFlow was set to be 0.02 years. An even smaller time step (0.002 years) was used for the CrunchFlow simulation but no noticeable difference in results was observed. For the MATLAB routine, advection is evaluated every 0.2 years, a smaller time step is possible with longer computing time. The same *Darcy* velocity 0.01 $m^3/m^2$/yr (or 0.0147 m/yr for fluid velocity assuming 0.68 porosity) was used in both simulations. Two cases were simulated: a fluid with high concentration of chloride advects downward (Figure S2A) and a high chloride concentration pulse being transported by the advected fluid (Figure S2B).

It is obvious from Figure S2A that the advection simulation done by CrunchFlow shows significant numerical dispersion as the edges of the square function are gradually smeared with time. On the contrary, the square functions were better preserved in the simulations done by the MTALAB routine. In the other case, where a pulse of Cl-rich fluid is transported by the advected fluid (Figure S2B), the peak heights in the simulations done by CrunchFlow were reduced by 9.7%, 18%, and 20% comparing to the original peak height. For the simulations done by MATLAB, the peak heights were only 3.6% reduced at most after simulating the flow for 140 years. The reduced peak height and gradually spreading peak are clear characteristics of numerical dispersion during the simulation of advection. This comparison justifies our decision not to use the built-in function of CrunchFlow to simulate advection. Our MATLAB routine provides a more accurate and numerically stable alternative for this purpose.

Figure S2: Comparison of fluid advection simulated by CrunchFlow built-in function and our MATLAB routine. Grey dashed lines and blue solid lines are the results from CrunchFlow and our MATLAB routine, respectively. The red solid and dashed lines in (B) mark the initial concentration of Cl. As we included only fluid advection in both models, there should be no concentration reduction during fluid transport. The reduction between the model results and the initial Cl concentration, as shown by the percentages (the blue arrow in (B)), is solely due to numerical dispersion in both models. Our MATLAB routine provides more accurate model results with less numerical dispersion.

[Figure]

---

## Author Comment (AC1) · 18 May 2016

We thank Dr. Sivan for her positive review and the compliment. We reply to the comments below and aim to revise the manuscript accordingly.

Referee's comment: The model is not detailed and explained enough. You should cite less Rooze et al 2016 and provide more details here. Also, you should perform sensitivity tests for the various uncertainties. I did not have access to Rooze et al paper, but from its title I am assuming it is not on the same system so there is no overlapping. You should however upload this paper.

Author's reply: We will expand the model description in the revised version of the

manuscript. In addition, we will perform appropriate sensitivity tests to demonstrate the various uncertainties (see also reply to referees #2 and #3). The study by Rooze et al. can be at http://onlinelibrary.wiley.com/doi/10.1002/lno.10275/abstract

Referee's comment: The $Fe_{2+}$ increase in the deep sediments could be from deep Fe-AOM as we see in lakes and coastal sediments (Sivan et al., 2011; Egger et al., 2015), however it can also be organoclastic. There may be reactivation of less soluble Fe(III) minerals in this system by other means other than methane oxidation (e.g as described by Sivan et al., 2016). You indeed mention it, however, you should refer to it as a possible option.

Author's reply: We will clarify that the reactivation of more crystalline Fe oxides by methanogens as described by Sivan et al. (2016) represents a possible mechanism for the Fe reduction at depth in the sediment.

Referee's comment: The assumption that the total dissolved Fe and Mn (as measured by AE-ICP) are Fe(II) and Mn(II) is probably fine, however you should test it and show it in at least in one of the profiles in the Black sea sediments (or cite other works that did it there). You should compare the Fe(II) to Fe(total) by another method (as the Ferrozine), or compare your assumed Fe(II) to Fe(II) from the Ferrozine or another method.

Author's reply: Unfortunately, we cannot test this as we acidified our pore water samples prior to analysis for Fe and Mn with ICP-OES. Although the solubility of $Fe(OH)_3$ is very low at natural pH and the occurrence of dissolved $Fe_{3+}$ is highly unlikely in reducing sediments, $Fe(OH)_3$ or $Fe(OH)_{2+}$ complexes could pass through the 0.45 $\mu$m filters used in this study (Raiswell and Canfield, 2012; Geochem. Perspect.). In addition, a small fraction of the dissolved Mn could indeed be present as $Mn_{3+}$, as shown for suboxic surface sediments (Madison et al., 2013; Science). We will clarify that the dissolved Fe and Mn in our study refers to Fe and Mn passing through a 0.45 $\mu$m filter and thus likely consists of a mixture of truly dissolved (aqueous), as well as organically

complexed, colloidal and nanoparticulate Fe and Mn species in the method section of the revised manuscript.

Referee's comment: You should discuss in more detail the sulfate profile and – its apparent "diffusion" profile (linear curve) with organoclastic sulfate reduction, and the cryptic S cycle in the upper part of this profile. You should also compare the downward flux of sulfate to the SMTZ and the upward flux that you calculated for methane and discuss it.

Author's reply: As suggested by the referee, we will discuss the sulfate profile in more detail and also elaborate on the relative fluxes of sulfate and methane to the SMTZ.

Referee's comment: The $\delta$13C of methane similarity to Yoshinaga's data is convincing and satisfactory. Interpretation/speculations regarding the profile of $\delta$D of methane should be given.

Author's reply: The profile of $\delta$D-CH4 will be discussed in more detail in the revised manuscript.

Referee's comment: L 84-85: Vivanite was found also in Lake Kinneret (Sivan et al., 2011), it can support your finding and related processes.

Author's reply: We will add this additional reference in the text.

Referee's comment: L 92-93: Also propose the other option for Fe reduction.

Author's reply: Reactivation of more crystalline Fe oxides by methanogens will be added as an additional option (see comment above).

Referee's comment: L 141-142: See comment regarding this method above.

Author's reply: We will revise these lines according to the comment above.

Referee's comment: L. 151: I assume the auto analyzer was based on IR. How did you remove of the sulfide?

Author's reply: Samples for HPO42-, DIC and HS- were analyzed colorimentrically on two separate QuAAtro (SEAL Analytical, Germany) auto-analyzers in thermo-stated containers. HPO42- was measured at 880 nm after the formation of molybdophosphate-complexes (Murphy and Riley, 1962; Anal. Chim. Acta). Samples for DIC were acidified online after being oxidized by H2O2 to prevent H2S loss and analyzed as described by Stoll et al. (2001; Anal. Chem.). The sulfide was trapped with NaOH and analyzed using the methylene blue method as described by Grasshof (1969; Woods Hole Oceanographic Institute). We will add these details in the manuscript.

Referee's comment: L. 236-242: Clarify and explain this part in more details.

Author's reply: We will expand this paragraph in the revised manuscript.

Referee's comment: L. 288: Show how you calculated to this saturation value and under which salinity conditions. Mark this value on the figures of methane too.

Author's reply: We will provide information about how this saturation concentration was derived and, as suggested by the referee, we will indicate the saturation concentration of CH4 under atmospheric conditions in the Figures.

Referee's comment: Add the bottom water values on the porewater profiles

Author's reply: The bottom water values are already given in the pore water profiles (depth of 0 cm).

Referee's comment: L 297: Change the sentence to a more precise one.

Author's reply: This sentence will be improved in the revised version.

Referee's comment: L. 381: Explain the other 91% based on the profile (see main comment).

Author's reply: We assume the referee means 81 %. We will add the explanation in the revised manuscript.

Referee's comment: L 445-451: See the main comment regarding organoclastic Fe reduction.

Author's reply: We will expand this section and discuss why we assumed Fe-AOM as the only Fe reduction pathway in the model (also see comments to referee # 2).

Referee's comment: L. 566-568: You don't need this trivial sentence, your work is good and nice enough without it.

Author's reply: We thank the referee for the compliment, but prefer to keep this sentence. In our opinion, it is important to include this statement because such diagenetic redistributions are often not considered in paleoceanographic studies.

Referee's comment: Fig. 3: No sulfate measurements in the sapropel depths ? Add saturation of methane. What could be the reaction precipitating phosphate in the upper 300 cm (hydroxyl-apatite)?

Author's reply: Unfortunately, the pore water samples for sulfate measurements in the sapropel depths were lost prior to analysis. We will provide information about the $CH_4$ saturation concentration in the revised version. We suggest that the removal of dissolved phosphate in the upper 300 cm of sediment is due to authigenic apatite formation as observed previously in sediments of the Black Sea (Dijkstra et al. 2014; PlosONE). We will add this information to the revised manuscript.

Referee's comment: Fig. 4: Again with the saturation of methane.

Author's reply: See comment above.

Referee's comment: L 339: Start a new subchapter.

Author's reply: We will add a subchapter for the rates and temporal evolution.

Sincerely, on behalf of all authors,

Matthias Egger

---

## Author Comment (AC2) · 18 May 2016

We thank Dr. Hong for encouraging and critical remarks. We reply to the comments below and aim to revise the manuscript accordingly.

Referee's comment: Major comments Modeling numerical issue: In conventional models for transportreaction models, advection (i.e. sediment burial) often inevitably results in numerical dispersion, concentration will decrease as time progresses with burial even without any reaction. This effect will be especially obvious when using high advection rate (burial rate), large time discretization, and a long modeling time. I've done some tests before (Hong et al., 2016 accepted by limnology and oceanography) by simulating time progression of a profile with sharp concentration change. After 140

years of simulation, the concentration is 20% reduced compared to the value it should have (see the attached file for this comparison). As for the sharp increase of OM content in your environment, you will inevitably encounter this numerical issue. I urge the authors to run some simulations with only burial (no diffusion and other reactions), and see how your sediment and porewater profiles will progress.

Author's reply: We thank the referee for sharing his experience with us. The ReacTran package applied in this study (see line 261) accounts for numerical dispersion using total variation diminishing (TVD) slope limiters. A description of the ReacTran package can be found at: https://cran.r-project.org/web/packages/ReacTran/ReacTran.pdf

Referee's comment: I also wonder what is the consequence to accelerate your model. The price of numerical issues will be greater when you accelerate it by using larger time steps and/or faster rates. It is almost no way to have a model that is both efficient and accurate. There is always a sacrifice.

Author's reply: We could not find any indication that the model acceleration impacts the conclusions presented in this study. A higher sedimentation rate in the lake phase, however, largely improves the model efficiency as it significantly reduces the computing time.

Referee's comment: The other potential numerical issue I want to point out for the authors is the convergence of the model results. You have to make sure you use temporal and spatial steps small enough so that the results are stable. This can be done by running the model several times (the same reactions and setup) with smaller time/space discretization for each run. Chose the smallest discretization that your model results stop changing.

Author's reply: We used this approach in this study (i.e. we ran the model using various time and space discretizations). The model presented here thus reflects the smallest discretization for which the model results stop changing.

[Figure]

Referee's comment: Conflict between observations and model results: In a few places in the paper, the authors didn't explain clearly the conflict between observations and model results. One example is the choice of chloride changes with time. The authors used a very different evolution pattern from what literature suggested because it provides a better fit of their chloride concentration. However, the time scale adopted by the authors (100yrs) is an order smaller than what is suggested in the literature (2000yrs). The authors provide no explanation about such difference. I envision that if the author use constant chloride from 2000yrs BP until now and increase fluid advection rate (larger u), they might be able to fit the profile. I think the authors should explain better why choosing such condition.

Author's reply: We will elaborate on the salinization scenario used in this study and provide potential explanations for the offset with the literature.

Referee's comment: The other example is from line 369 to 371. The authors claimed the SR rates they estimated from the model in zone I & II are more correct the estimation from porewater profiles. This statement raises the question that, then how do you know the SR rate you estimated from these two zones are accurate since you have no data to support you.

Author's reply: Sulfate reduction rates derived from pore water profiles represent net sulfate consumption, while rates derived from radiotracer injection and diagenetic modeling reveal total sulfate turnover in the sediment, i.e. gross sulfate reduction rates (e.g. Jørgensen (1978; Geomicrobiol. J.) and Jørgensen et al. (2001; Deep Sea Res. Part I)). Thus, modeled rates are generally higher than rates estimated from pore water profiles. The referee is further kindly referred to lines 365-366 where we provide references to studies that have measured sulfate reduction rates in Black Sea sediments, which compare well with our model results.

Referee's comment: Very high methanogenesis rate in sulfate reduction zone: In fig 6, there are two peaks of methane production (one in bottom water or first cm of

sediment? While the other in zone II). My questions are two: 1) It is apparent that this methanogenesis is from OM decomposition. However, methanegenesis should be suppressed when the sulfate content is high, as in the case of zone II. I understand that although methanogenesis is inhibited by sulfate content (E5 & E6 in Table 4), model can still produce very high ME rate when there is ultrahigh OM content. However, a model is a model, do you have any prove such high methanegenesis from your zone I and II. Considering the CH4 production rate and SO4-AOM rate from Fig.6. you should see either high methane or light d13C of methane in zone I and II if the rate is this high. I however don't quite see those from your profiles.

Author's reply: The CH4 production rate depicted in Fig. 6 is not correct, as it should be multiplied with the solid volume fraction of the sediment (increases from $\sim 0.05$ at the sediment surface to $\sim 0.39$ at depth) and divided by a factor of two (only 0.5 mole of CH4 are produced per mole of organic matter during methanogenesis; see reaction R5 in Table 3; also see response to referee #2). Actual modeled rates of CH4 production are thus < 30 pmol cm-3 d-1 in the marine deposits, which is an order of magnitude lower than modeled SO4-AOM rates. Our modeled CH4 production rates in the surface sediments are higher compared to previous measurements from the Black Sea (e.g. Knab et al., 2009; Biogeosciences), but still significantly lower than net rates of methanogenesis measured in surface sediments of the Peruvian margin of up to $\sim$ 1 nmol cm-3 d-1 in the sulfate reduction zone (Maltby et al., 2016; Biogeosciences). We will add this information to the revised manuscript.

Referee's comment: 2) Back to fig6, your rates do not seem to balance. The highest CH4 production rate approaches 300 pmol/cm3/d which only stimulates an AOM rate less than maybe 20 pmol/cm3/d. If there is more production than consumption, isn't that you will methane accumulates in the porewater (i.e., high methane from that depth in the sediments). SR rate is over 2000 pmol/cm3/d in this section but sulfide production is only 300 pmol/cm3/d. where is the rest of sulfide production?

Author's reply: We thank the referee for pointing out these inconsistencies. There

has been an error in the plotting of the rates (see also answer to referee #2). The rate plots of Fig. 6 were not normalized to the same volume, meaning that the SO4 reduction rate, CH4 production and S0 disproportionation should be multiplied by the solid volume fraction of the sediment, while rates of SO4-AOM and Fe-AOM should be multiplied by the sediment porosity. In addition, only 0.5 mole of CH4 are produced per mole of organic matter during methanogenesis (see reaction R5 in Table 3). We will carefully check the unit conversion in the plotting, as well as the mass balance and revise Fig. 6 accordingly.

Referee's comment: The very complicated model: The authors use a rather complicated model in this study by choosing many reactions that are not totally necessary. For example, the authors choose to include aerobic processes (R1, R7-R12) and nitrate reduction (R2) even though there is no constraints on O2 and nitrate content in the porewater. I would also doubt the importance of these reactions due to the anoxic bottom water in Black Sea. The authors chose not to include Mn reduction due to its low content, which is fine with me, but decide to include all other processes that cannot be constrained? That is an odd decision to me. By excluding these unnecessary reactions, the authors can also improve the efficiency of the model.

Author's reply: Oxic mineralization and nitrate reduction were implemented because of the oxic Black Sea lake phase. Bottom water concentrations of oxygen and nitrate for the Lake phase were taken from Reed et al. (2011; GCA). We agree with the referee that nitrate reduction plays only a minor role (also during the oxic Lake phase), but prefer to keep the reaction in the model, as removing it does not significantly improve the model efficiency.

Referee's comment: I also wonder, with all the reactions assigned in the model, do the authors have enough constraints? I believe the answer should be close to yes as the authors have many data to support the model (which is very nice). I would urge the authors to spare a section in the text discussing the constraints for the model. To me, this is an extremely important but often ignored aspect in papers like this. I have done

some initial analyses based on the reaction network in Table 3. For example, for Fe2+, the authors have R3, R9, R10, R13, R14, and R20 for sources, and R8, R15, R23, and R24 for sinks. Some source and sink terms may be constrained by the data of iron mineral speciation. When the same analyses being applied on HS in porewater, it seems like the abundance of different Fe-S minerals also depend on the source and sink terms of HS. A table such as tab6 but with more species included may be useful for such discussion.

Author's reply: We kindly refer the referee to lines 229-231 and Table 5 where we describe the parameter constraints. We will add a table with the mass balance for each species in the Supplementary Information.

Referee's comment: One last comment on the complicated model, how does the model describe pH, which should be very important determining the type of dissolved sulfide and DIC. I don't see reactions such as H2S becomes HS-+H+ in Tab3 which describe the buffer capability of HS species (need same reactions for carbonate systems). Although there is usually no good constraint on pH, it's good to make sure pH falls in the right range especially when including pH-sensitive reactions.

Author's reply: The model does not include pH, because it does not capture the precise underlying reaction mechanisms. Furthermore, we do not have pH data to compare the model results to. Adding pH would be a separate study in itself.

Referee's comment: Minor comments Line 151: Please specify how you measure sulfide, phosphate, and DIC onboard.

Author's reply: We will add this information in the revised manuscript (see reply to referee #1).

Referee's comment: Line 211: why 20 meters? You should mark you explain this in the supplemental material.

Author's reply: The referee is kindly referred to lines 257-258 and Supplementary Fig.

S4.

Line 255: is zero gradient a good assumption for methane? How do you know there is no deeper source of methane?

Author's reply: We have no information about a potential deep source of CH4 at our study site. However, the good fit between the modeled and measured ammonium profile (Fig. 3) indicates that it is likely that most of the CH4 is produced within the model domain. We therefore think that a zero gradient assumption is a reasonable assumption for CH4.

Referee's comment: Line 289 to 293: You have same ammonium but higher methane in site4 and 5. Of course more severe degassing during core recovery in site 4 can be one explanation, but maybe there is more methane input from site 5 from greater depth. This echoes back my previous comment: is zero gradient really a good assumption for methane?

Author's reply: If the difference in the CH4 profiles between site 4 and 5 would be due to more CH4 input from greater depth at site 5, it would imply that the measured concentrations at site 4 represent actual in-situ concentrations of pore water CH4. However, we were not able to reproduce the observed ammonium profiles with such low rates of methanogenesis at depth. In contrast, our model suggests that in order to have sufficient pore water ammonium, CH4 concentrations should be significantly higher than the measured concentrations at both sites. The high CH4 concentrations derived from the model are also consistent with our observations of massive CH4 degassing during coring and with a previous observations in the western Black Sea shelf (Jørgensen et al., 2001; Deep Sea Res. Part I). We conclude that the ammonium profiles indicate that most of the CH4 is produced within the model domain, rather than from greater depths, thus supporting our zero gradient assumption.

Referee's comment: Line 304: How do you know the isotopic signature of methane is not affected by degassing?

Author's reply: We base this conclusion on the smooth pore water profiles of d13C-CH4 and dD-CH4 (see Fig. 4). To date, little is known about potential isotopic fractionation during degassing. However, it is thought that fast degassing is unlikely to result in major isotopic fractionation, as all CH4 isotopes are lost simultaneously. We will clarify this sentence in the revised manuscript.

Referee's comment: Line 533: Isn't that this will be capture in your orgP analyses?

Author's reply: Mineral formation in microbial cells is not included in the model. Inclusions of Fe-P minerals initially formed in bacteria could further also still come out in the CDB step of the SEDEX extraction.

Referee's comment: Line 827: Do you have any constrain on C/P ratio? Maybe this explains why the fitting on porewater phosphate profile is not as good?

Author's reply: The C/P ratio observed in the sedimentary record does not necessarily directly relate to the initial C/P ratio of the organic matter deposited on the seafloor due to the preferential regeneration of phosphorus from organic matter during anaerobic degradation (e.g. Ingall et al., 1993; GCA). It is thus of limited use as a constraint in this study.

Referee's comment: Salinity/chloride: In many places of the paper, the authors mixed the term salinity and chloride concentration (e.g., line268-282). Of course these two properties are usually linear dependent on each other but they are fundamentally different and may correlate with each other very differently when Black Sea was more of a "lake" or a "Sea". I suggest the authors to use chloride concentration throughout the paper or explain how they convert salinity to chloride concentration.

Author's reply: We will follow the referee's suggestion and explain how salinity is converted to chloride concentration in the revised manuscript.

Referee's comment: FigS3: What is going on with the very high alkalinity at very top? where dic concentration looks normal...

[Figure]

Author's reply: We thank the reviewer for pointing this out. We checked the raw data and conclude that the alkalinity data of the multicore at site 4 should not be trusted due to uncertainty over data quality. We will thus remove this data from the Supplementary Information.

Sincerely, on behalf of all authors,

Matthias Egger

---

## Author Comment (AC3) · 18 May 2016

We thank the anonymous referee for the positive remarks. We reply to the comments below and aim to revise the manuscript accordingly (i.e. we will include sensitivity analyses and will clarify our model formulation and corresponding results further).

Referee's comment: It is particularly the conclusions concerning sulfate- and iron-coupled AOM that require attention. The occurrence of Fe-AOM appears to be forced by the exclusion of organoclastic Fe reduction from the model, although there is plenty of evidence that organotrophic microbes can reduce crystalline Fe oxides, and there is no evidence that organotrophic Fe reduction cannot co-occur with methanogenesis if Fe reduction is limited by the availability/reactivity of iron oxides. Furthermore, is

seems that partitioning of AOM must be sensitive to the parameterization of the pathways, which therefore needs to be discussed.

Author's reply: We acknowledge that the reason for the exclusion of organoclastic Fe reduction at depth in the sediment requires a more detailed explanation. The reason for this assumption was to test how important Fe-AOM could be for the CH4 cycle, assuming that all the Fe reduction at depth would be exclusively due to Fe-AOM. We agree with the referee that organoclastic Fe reduction also provides a possible Fe reduction pathway and also state this in the manuscript (see, for example lines 31-33, 476-478 and 559-560). In the revised manuscript, we will clarify our model results and formulation, as well as include an appropriate sensitivity analysis.

Referee's comment: 22-23+89-90: The finding that sulfate-AOM enhances the sulfide flux is not novel according to lines 72-75.

Author's reply: We agree with the referee, and we will rephrase the text so that it is clear that we are not implying that this is a novel finding.

Referee's comment: 289-96: Just a comment: The difference in the two methane profiles is strange and it is difficult to understand how degassing would have caused a proportional decrease in methane in the zone above the zone of saturation. Nonetheless, I agree that it is the most likely explanation given the similarity of all other profiles, including the methane isotopes. I suggest rephrasing 293-294 to clarify which methane data this applies to.

Author's reply: We will elaborate on the CH4 degassing and clarify our conclusions (lines 293-294) in the revised manuscript.

Referee's comment: 339-41+Fig 6: I don't understand the very high rates of sulfate reduction and methanogenesis in the sapropel, and the model doesn't seem to fit the data well here. Albeit noisy, the measured H2S profile seems straight or even concave in this region, and the same clearly goes for DIC, whereas the model profiles are

convex, which suggests that the model overestimates the rates substantially. Although this zone is not of primary interest in this study, an overestimation of rates and product concentrations results in a shallower gradient from unit II to the SMTZ and therefore in lower sulfate-AOM rates, so the fit here still influences the central conclusions.

Author's reply: We will work on improving the model fits for H2S and DIC in the marine deposits and re-evaluate the modeled rates.

Referee's comment: Fig. 6, further+ Fig. 3: There seems to be an error in the H2S production panel in Fig. 6 as H2S production from sulfate reduction is only a fraction of the sulfate reduction rate? Shouldn't hese be 1:1 as is the case for sulfate-AOM and sulfide production from sulfate AOM? Also, what happens to methane produced in unit II? The methane profile appears flat, yet only a fraction of the production is consumed by AOM. Please provide blow-ups of modelled methane in the upper 2 m and of sulfate below the SMTZ in Fig. 3.

Author's reply: We thank the referee for pointing out these inconsistencies. The rate plots of Fig. 6 were not normalized to the same volume, meaning that the SO4 reduction rate, CH4 production and S0 disproportionation should be multiplied by the solid volume fraction of the sediment, while rates of SO4-AOM and Fe-AOM should be multiplied by the sediment porosity. In addition, only 0.5 mole of CH4 are produced per mole of organic matter during methanogenesis (see reaction R5 in Table 3). We will carefully check the unit conversion in the plotting, as well as the mass balance and revise Fig. 6 accordingly.

Referee's comment: 391-5: This is the only real flaw in the paper. The Rayleigh function applies to closed systems and should never be used in open systems such as this one, where diffusion affects the relative distribution of the isotopes. Accurate enrichment factors can only be derived through modelling (e.g., Alperin et al. 1988). The closed-system approach will underestimate enrichment factors substantially in most cases, and likely explains the low value derived here. This problem was described

decades ago (e.g., Jørgensen 1979, GCA 43:363).

Author's reply: We thank the referee for this critical remark. We agree that modeling of the methane isotopes would result in a more reliable estimation of the enrichment factors. However, isotopic fractionation was not the main aim of this study. In addition, considering that diffusion fractionation is likely minor compared to the fractionation associated with oxidation (e.g. Happell et al., 1995; Limnol. Oceanogr.; Chanton, 2005; Org. Geochem) and that diffusion might be slower than oxidation in our settings, it could be argued that we may be looking at a quasi close environment. We will clarify the limitation of our approach in the revised manuscript.

Referee's comment: 401: I think some of these studies observed sulfate reduction and did not only postulate it?

Author's reply: We agree with the referee and will clarify this in the revised version.

Referee's comment: 442-3: Under which conditions, if any, within a realistic parameter space or with an alternative set of reactions, would a cryptic sulfur cycle be able to explain the accumulation of Fe2+?

Author's reply: We will elaborate on this in the revised manuscript.

Referee's comment: 450-5: The references listed here suggest that AOM may be coupled to Fe reduction, but here you really use them to support the assumption that Fe reduction can be coupled to AOM rather than to organoclastic Fe reduction – Is there any support for that in any of those references? As stated in l. 463, organoclastic Fe reduction is clearly limited at these depths, but that doesn't mean that it is absent. Furthermore l. 474-6 seems to suggest organoclastic Fe reduction anyway, even if it is by archaea? But what special skills do these organisms have that would enable them to reactivate Fe oxides?

Author's reply: We acknowledge that lines 450-455 could be formulated better. The existing literature cited here indicates that Fe reduction could be coupled to CH4 oxidation in aquatic sediments. The aim of this study was thus to evaluate whether the geochemical profiles could be reproduced assuming that all Fe reduction at depth would be coupled to Fe-AOM and to show the potential impact of Fe-AOM on the CH4 cycle. We are not claiming that Fe-AOM is more likely than organoclastic Fe reduction, but rather show that Fe-AOM represents a plausible mechanism for the deep Fe reduction. We will clarify this in the revised manuscript. To our understanding, the underlying mechanisms of the reactivation of Fe oxides by methanogens as described by Sivan et al. (2016; Geobiology) remain enigmatic.

Referee's comment: 489-93: It seems trivial that in situ rates under the given conditions are low compared to lab-based rates. What is the observational basis for the parameterization of the reaction?

Author's reply: The Fe-AOM reaction was implemented according to Beal et al. (2009; Science) and the rates were derived by fitting the modeled pore water and solid phase profiles with the observations as explained in the text (lines 229-231). In our view, it is not trivial to mention the difference in rates calculated here and observed in laboratory experiments in other settings. It illustrates that it may be difficult to perform laboratory incubations of Black Sea sediments to study Fe-AOM.

Referee's comment: 494-5: How sensitive is the sulfate/Fe-AOM partitioning to the parameterization?

Author's reply: We will perform an appropriate model sensitivity analysis in the revised manuscript (see earlier comments).

Referee's comment: Table 3, R6+16: I understand that you need a sink for H2, but why is it only methanogenesis and not, at least sulfate reduction? This will lead to overestimation of methanogenesis in the sulfate zone.

Author's reply: H2 is not included in our model.

Referee's comment: Table 4: R19+ R20 are biological processes and as such might

obey biological (saturation) kinetics? These are key reactions in the paper and the observational basis for the kinetic expressions, and their impact on the conclusions should be discussed.

Author's reply: The bimolecular rate equation expression for AOM applied in this study is the most frequently used AOM parameterization in reactive transport models (e.g. Regnier et al., 2011; Earth-Science Reviews). We followed the bimolecular approach because of the high uncertainty in half-saturation constants, in particular for the putative Fe-AOM pathway. For a detailed discussion about AOM parameterization, the referee is kindly referred to the study by Regnier et al. (2011; Earth-Science Reviews).

Referee's comment: Table 6 + Fig 6: The labelling of the two kinds of methanogenesis is misleading. The light isotopic composition of methane implies that it is formed mainly through CO2 reduction rather than acetoclastic methanogenesis, i.e. that "Methanogenesis (OM)" is mainly CO2-based. "Methanogenesis (DIC)" is really a peculiarity of the model and completely and uniquely linked to pyrite formation, so "Methanogenesis (FeS2)" would be more appropriate (but see also comment to Table 3 above).

Author's reply: We agree with referee that reaction R6 in Table 3 and the labeling in Table 6 and Fig. 6 are difficult to understand and will improve it in the revised version.

Referee's comment: Fig. 7: Consider a colour version here. The darkest shading on the scale bars always appears darker than the darkest part of the figures. Because the shading varies so little from min. to max. it is very difficult to extract quantitative information.

Author's reply: We will change Fig. 7 to a color version.

Sincerely, on behalf of all authors,

Matthias Egger

---

## Author Response (AR1)

**Universiteit Utrecht**

**Faculty of Geosciences**

Department of Earth Sciences

**Telephone**: +31 (0)30 253 50 05

Biogeosciences

**E-mail**: m.j.egger@uu.nl

**Date: June 24, 2016**

**Subject: Submission of revised manuscript bg-2016-64**

Dear Tina,

Hereby we submit the revised manuscript bg-2016-64 **"Anaerobic oxidation of methane alters sediment records of sulfur, iron and phosphorus in the Black Sea"** by Matthias Egger, Peter Kraal, Tom Jilbert, Fatimah Sulu-Gambari, Célia J. Sapart, Thomas Röckmann and Caroline P. Slomp.

We would like to thank the three reviewers for their encouraging and critical remarks that helped to improve our manuscript. With this letter, we have also added a detailed reply to the reviewer's comments including an explanation of how and where each point raised by the reviewers was incorporated in the manuscript, a list of all relevant changes made in the manuscript, as well as a marked-up manuscript version. We followed the reviewer's recommendations in most cases.

Please note that all line numbers in our replies refer to the revised text, unless stated otherwise.

Sincerely, on behalf of all authors,

**Matthias Egger**

**Response to reviewers for manuscript bg-2016-64**

**Referee # 1 (Orit Sivan)**

*Referee's comment: This paper deals with diagenetic processes in the sediments of the Black Sea which changed from a lacustrine environment to a marine system. The work focuses on AOM and its effect on the linked species and processes under these changes. This was done by producing solid phase and porewater profiles, and by diagenetic modeling. The work is well written and easy to read, and I found it complete, serious and convincing. The authors measured, calculated and thought on almost all the possible aspects that could affect this system during these changes. This careful work enhances our understanding on AOM by iron and sulfate in marine setting in general and specifically in a complex setting. It also provides us with new knowledge on the Black Sea's limnological history. I thus suggest accepting this work pending minor comments.*

Author's reply: We thank the referee for the compliment. We reply to the comments below and have now revised the manuscript accordingly.

*Referee's comment: The model is not detailed and explained enough. You should cite less Rooze et al 2016 and provide more details here. Also, you should perform sensitivity tests for the various uncertainties. I did not have access to Rooze et al paper, but from its title I am assuming it is not on the same system so there is no overlapping. You should however upload this paper.*

Author's reply: We have expanded the model description in the revised version of the manuscript **(lines 247-257)**. In addition, we have performed appropriate sensitivity tests and show the uncertainties associated with key processes in the Supplementary Information (see also reply to referees #2 and #3) **(lines 241-243 and Supplementary Figs. S2 and S3)**. Note that we changed the model grid from constant to exponentially decreasing **(lines 280-282)**. This change had no effect on the results, but helped to speed up the sensitivity analyses. The study by Rooze et al. can be found at http://onlinelibrary.wiley.com/doi/10.1002/lno.10275/abstract

*Referee's comment: The Fe2+ increase in the deep sediments could be from deep Fe-AOM as we see in lakes and coastal sediments (Sivan et al., 2011; Egger et al., 2015), however it can also be organoclastic. There may be reactivation of less soluble Fe(III) minerals in this system by other means other than methane oxidation (e.g as described by Sivan et al., 2016). You indeed mention it, however, you should refer to it as a possible option.*

Author's reply: We have clarified that the reactivation of more crystalline Fe oxides by methanogens as described by Sivan et al. (2016) represents a possible mechanism for the Fe reduction at depth in the sediment **(lines 32; 95-96; 606-607 and 1022-1023)**.

*Referee's comment: The assumption that the total dissolved Fe and Mn (as measured by AE-ICP) are Fe(II) and Mn(II) is probably fine, however you should test it and show it in at least in one of the profiles in the Black sea sediments (or cite other works that did it there). You should compare*

*the Fe(II) to Fe(total) by another method (as the Ferrozine), or compare your assumed Fe(II) to Fe(II) from the Ferrozine or another method.*

Author's reply: Unfortunately, we cannot test this as we acidified our pore water samples prior to analysis for Fe and Mn with ICP-OES. Although the solubility of $Fe(OH)_3$ is very low at natural pH and the occurrence of dissolved Fe3+ is highly unlikely in reducing sediments, $Fe(OH)_3$ or $Fe(OH)^{2+}$ complexes could pass through the 0.45 µm filters used in this study (Raiswell and Canfield, 2012; *Geochem. Perspect.*). In addition, a small fraction of the dissolved Mn could indeed be present as Mn3+, as shown for suboxic surface sediments (Madison et al., 2013; *Science*). We have now included that the dissolved Fe and Mn in our study refers to Fe and Mn passing through a 0.45 µm filter and thus likely consists of a mixture of truly dissolved (aqueous), as well as organically complexed, colloidal and nanoparticulate Fe and Mn species in the method section of the revised manuscript **(lines 146-148)**.

*Referee's comment: You should discuss in more detail the sulfate profile and – its apparent "diffusion" profile (linear curve) with organoclastic sulfate reduction, and the cryptic S cycle in the upper part of this profile. You should also compare the downward flux of sulfate to the SMTZ and the upward flux that you calculated for methane and discuss it.*

Author's reply: As suggested by the referee, we now discuss the sulfate profile in more detail **(lines 397-404)** and also elaborate on the relative fluxes of sulfate and methane to the SMTZ **(lines 414-418)**.

*Referee's comment: The δ13C of methane similarity to Yoshinaga's data is convincing and satisfactory. Interpretation/speculations regarding the profile of δD of methane should be given.*

Author's reply: The profile of $\delta D\text{-}CH_4$ is now discussed in more detail in the revised manuscript **(lines 424-427)**.

*Referee's comment: L 84-85: Vivanite was found also in Lake Kinneret (Sivan et al., 2011), it can support your finding and related processes.*

Author's reply: We have added this additional reference in the text **(line 86)**.

*Referee's comment: L 92-93: Also propose the other option for Fe reduction.*

Author's reply: Reactivation of more crystalline Fe oxides by methanogens has been added as an additional option (see comment above).

*Referee's comment: L 141-142: See comment regarding this method above.*

Author's reply: We have revised these lines according to the comment above.

*Referee's comment: L. 151: I assume the auto analyzer was based on IR. How did you remove of the sulfide?*

Author's reply: Samples for $HPO_4^{2-}$, DIC and $HS^-$ were analyzed colorimentrically on two separate QuAAtro (SEAL Analytical, Germany) auto-analyzers in thermo-stated containers. $HPO_4^{2-}$ was measured at 880 nm after the formation of molybdophosphate-complexes (Murphy and Riley, 1962; *Anal. Chim. Acta*). Samples for DIC were acidified online after being oxidized by $H_2O_2$ and analyzed as described by Stoll et al. (2001; *Anal. Chem.*). The sulfide was trapped with NaOH and analyzed using the methylene blue method as described by Grasshof (1969; *Woods Hole Oceanographic Institute*). We have now added these details in the manuscript **(lines 157-162)**.

*Referee's comment: L. 236-242: Clarify and explain this part in more details.*

Author's reply: We have expanded this paragraph in the revised manuscript **(lines 252-257)**.

*Referee's comment: L. 288: Show how you calculated to this saturation value and under which salinity conditions. Mark this value on the figures of methane too.*

Author's reply: We have provided information on how this saturation concentration was derived **(lines 311-312)** and, as suggested by the referee, we have indicated the saturation concentration of CH4 under atmospheric conditions in the Figures **(see Figs. 3, 4 and S5)**.

*Referee's comment: Add the bottom water values on the porewater profiles*

Author's reply: The bottom water values are already given in the pore water profiles (depth of 0 cm).

*Referee's comment: L 297: Change the sentence to a more precise one.*

Author's reply: This sentence has been improved in the revised version **(lines 320-322)**.

*Referee's comment: L. 381: Explain the other 91% based on the profile (see main comment).*

Author's reply: We have added the explanation in the revised manuscript **(lines 383-392)**.

*Referee's comment: L 445-451: See the main comment regarding organoclastic Fe reduction.*

Author's reply: We have expanded this section and now discuss why we assumed Fe-AOM as the only Fe reduction pathway in the model (also see comments to referee # 2) **(lines 496-499 and 247-251)**.

*Referee's comment: L. 566-568: You don't need this trivial sentence, your work is good and nice enough without it.*

Author's reply: We thank the referee for the compliment, but prefer to keep this sentence. In our opinion, it is important to include this statement because such diagenetic redistributions are often not considered in paleoceanographic studies.

*Referee's comment: Fig. 3: No sulfate measurements in the sapropel depths ? Add saturation of methane. What could be the reaction precipitating phosphate in the upper 300 cm (hydroxyl-apatite)?*

Author's reply: Unfortunately, the pore water samples for sulfate measurements in the sapropel depths were lost prior to analysis. We now provide information about the $CH_4$ saturation concentration in the revised version **(see Figs. 3, 4 and S5)**. We suggest that the removal of dissolved phosphate in the upper 300 cm of sediment is due to authigenic apatite formation as observed previously in sediments of the Black Sea (Dijkstra et al. 2014; *PlosONE*) and have added this information to the revised manuscript **(lines 576-578)**.

*Referee's comment: Fig. 4: Again with the saturation of methane.*

Author's reply: See comment above.

*Referee's comment: L 339: Start a new subchapter.*

Author's reply: We have added a subchapter for the rates and temporal evolution **(line 372)**.

**Referee # 2 (anonymous)**

*Referee's comment: This study addresses the diagenetic implications of anaerobic methane oxidation in Black Sea sediments where marine deposits overlie a freshwater facies into which a sulfate front is advancing. High-resolution geochemical profiles of dissolved and solid species are presented from two adjacent sites, and the profiles are simulated in a complex non-steady-state diagenetic model that derives rates of the relevant processes. The subject is interesting, obviously relevant to Biogeosciences, and the results and conclusions presented here are novel and add substantially to our understanding of sediment biogeochemistry and diagenesis. The text is well written, clear, and concise, the data is good quality, and the conclusions are generally justified by the data and modelling. The authors particularly deserve credit for clearly distinguishing model results from reality. My main concern with the paper is that I miss a deeper analysis and discussion of the extent to which the modelling results are forced by the formulation and parameterization of the model. This could involve a sensitivity analysis or testing of alternative scenarios. Additionally, some aspects of the model results and formulation require clarification.*

Author's reply: We thank the referee for the compliment. We reply to the comments below and have revised the manuscript accordingly (i.e. we have included a sensitivity analysis in the Supplementary Information and have clarified our model formulation and the corresponding results further).

*Referee's comment: It is particularly the conclusions concerning sulfate- and iron-coupled AOM that require attention. The occurrence of Fe-AOM appears to be forced by the exclusion of organoclastic Fe reduction from the model, although there is plenty of evidence that organotrophic microbes can reduce crystalline Fe oxides, and there is no evidence that organotrophic Fe reduction cannot co-occur with methanogenesis if Fe reduction is limited by the availability/reactivity of iron oxides.*

Author's reply: We thank the referee for the positive remarks. We acknowledge that the reason for the exclusion of organoclastic Fe reduction at depth in the sediment requires a more detailed explanation. The reason for this assumption was that we wished to test how important Fe-AOM could be for the $CH_4$ cycle, assuming that all the Fe reduction at depth would be exclusively due to Fe-AOM. We agree with the referee that organoclastic Fe reduction also provides a possible Fe reduction pathway and also state this in the original manuscript (see, for example lines 31-33, 476-478 and 559-560). In the revised manuscript, we have clarified our assumption for Fe-AOM and corresponding model results and formulation **(lines 496-499 and 247-251)**.

*Referee's comment: Furthermore, is seems that partitioning of AOM must be sensitive to the parameterization of the pathways, which therefore needs to be discussed.*

Author's reply: The sensitivity of the AOM partitioning is now discussed in the caption of **Supplementary Fig. S2**.

*Referee's comment: 22-23+89-90: The finding that sulfate-AOM enhances the sulfide flux is not novel according to lines 72-75.*

Author's reply: We agree with the referee, and have rephrased the text so that it is clear that we are not implying that this is a novel finding **(lines 22 and 92-93)**.

*Referee's comment: 289-96: Just a comment: The difference in the two methane profiles is strange and it is difficult to understand how degassing would have caused a proportional decrease in methane in the zone above the zone of saturation. Nonetheless, I agree that it is the most likely explanation given the similarity of all other profiles, including the methane isotopes. I suggest rephrasing 293-294 to clarify which methane data this applies to.*

Author's reply: We elaborate on the degassing of $CH_4$ and clarify our conclusions in the revised manuscript **(lines 311-317)**.

*Referee's comment: 339-41+Fig 6: I don't understand the very high rates of sulfate reduction and methanogenesis in the sapropel, and the model doesn't seem to fit the data well here. Albeit noisy, the measured H2S profile seems straight or even concave in this region, and the same clearly goes for DIC, whereas the model profiles are convex, which suggests that the model overestimates the rates substantially. Although this zone is not of primary interest in this study, an overestimation of rates and product concentrations results in a shallower gradient from unit II to the SMTZ and therefore in lower sulfate-AOM rates, so the fit here still influences the central conclusions.*

Author's reply: The rates of sulfate reduction and methanognesis were corrected **(lines 363-367; see also Table 6 and Fig. 6)** and are now significantly smaller (see comment below). We have also improved the model fits for $H_2S$ and DIC in the marine deposits **(see Fig. 3)**. The fit still is not perfect, but the small offset does not impact our conclusions.

*Referee's comment: Fig. 6, further+ Fig. 3: There seems to be an error in the H2S production panel in Fig. 6 as H2S production from sulfate reduction is only a fraction of the sulfate reduction rate? Shouldn't hese be 1:1 as is the case for sulfate-AOM and sulfide production from sulfate AOM? Also, what happens to methane produced in unit II? The methane profile appears flat, yet only a fraction of the production is consumed by AOM. Please provide blow-ups of modelled methane in the upper 2 m and of sulfate below the SMTZ in Fig. 3.*

Author's reply: We thank the referee for pointing out these inconsistencies. The rate plots of Fig. 6 were not normalized to the same volume, meaning that the $SO_4$ reduction rate, $CH_4$ production and $S_0$ disproportionation should be multiplied by the solid volume fraction of the sediment, while rates of $SO_4$-AOM and Fe-AOM should be multiplied by the sediment porosity. In addition, only 0.5 mole of $CH_4$ are produced per mole of organic matter during methanogenesis (see reaction R5 in Table 3). We have changed the unit conversion in the plotting, as well as the mass balance and have revised **Fig. 6 and Table 6** accordingly.

*Referee's comment: 391-5: This is the only real flaw in the paper. The Rayleigh function applies to closed systems and should never be used in open systems such as this one, where diffusion affects the relative distribution of the isotopes. Accurate enrichment factors can only be derived through modelling (e.g., Alperin et al. 1988). The closed-system approach will underestimate enrichment factors substantially in most cases, and likely explains the low value derived here. This problem was described decades ago (e.g., Jørgensen 1979, GCA 43:363).*

Author's reply: We thank the referee for this critical remark. We agree that modeling of the methane isotopes would result in a more reliable estimation of the enrichment factors. However, the determination of isotopic fractionation was not the main aim of this study. In addition, considering that diffusion fractionation is likely minor compared to the fractionation associated with oxidation (e.g. Happell et al., 1995; *Limnol. Oceanogr.*; Chanton, 2005; *Org. Geochem*) and that diffusion might be slower than oxidation in our settings, it could be argued that we may be looking at a quasi-closed environment. We have now clarified the limitation of our approach in the revised manuscript **(lines 431-437)**.

*Referee's comment: 401: I think some of these studies observed sulfate reduction and did not only postulate it?*

Author's reply: We agree with the referee and have changed this in the revised manuscript **(line 443).**

*Referee's comment: 442-3: Under which conditions, if any, within a realistic parameter space or with an alternative set of reactions, would a cryptic sulfur cycle be able to explain the accumulation of Fe2+?*

Author's reply: We now elaborate on this in the revised manuscript **(lines 483-485)**.

*Referee's comment: 450-5: The references listed here suggest that AOM may be coupled to Fe reduction, but here you really use them to support the assumption that Fe reduction can be coupled to AOM rather than to organoclastic Fe reduction – Is there any support for that in any of those references? As stated in l. 463, organoclastic Fe reduction is clearly limited at these depths, but that doesn't mean that it is absent. Furthermore l. 474-6 seems to suggest organoclastic Fe reduction anyway, even if it is by archaea? But what special skills do these organisms have that would enable them to reactivate Fe oxides?*

Author's reply: We acknowledge that lines 450-5 in the original manuscript could be formulated better. The existing literature cited here indicates that Fe reduction could be coupled to $CH_4$ oxidation in aquatic sediments. The aim of this study was thus to evaluate whether the geochemical profiles could be reproduced assuming that all Fe reduction at depth would be coupled to Fe-AOM and to show the potential impact of Fe-AOM on the $CH_4$ cycle. We are not claiming that Fe-AOM is more likely than organoclastic Fe reduction, but rather show that Fe-AOM represents a plausible mechanism for the deep Fe reduction. We have now clarified this in the revised manuscript **(lines 496-499 and 247-251)**. To our understanding, the underlying mechanisms of the reactivation of Fe oxides by methanogens as described by Sivan et al. (2016; *Geobiology*) remain enigmatic.

*Referee's comment: 489-93: It seems trivial that in situ rates under the given conditions are low compared to lab-based rates. What is the observational basis for the parameterization of the reaction?*

Author's reply: The Fe-AOM reaction was implemented according to Beal et al. (2009; *Science*) and the rates were derived by fitting the modeled pore water and solid phase profiles with the observations as explained in the text (lines 239-241). In our view, it is not trivial to mention the difference in rates calculated here and observed in laboratory experiments in other settings. It illustrates that it may be difficult to perform laboratory incubations of Black Sea sediments to study Fe-AOM.

*Referee's comment: 494-5: How sensitive is the sulfate/Fe-AOM partitioning to the parameterization?*

Author's reply: We have performed an appropriate model sensitivity analysis in the revised manuscript and discuss its sensitivity in the caption of **Supplementary Fig. S2**.

*Referee's comment: Table 3, R6+16: I understand that you need a sink for H2, but why is it only methanogenesis and not, at least sulfate reduction? This will lead to overestimation of methanogenesis in the sulfate zone.*

Author's reply: The referee is kindly referred to Table 2, where it becomes evident that $H_2$ is not explicitly modeled in this study.

*Referee's comment: Table 4: R19+ R20 are biological processes and as such might obey biological (saturation) kinetics? These are key reactions in the paper and the observational basis for the kinetic expressions, and their impact on the conclusions should be discussed.*

Author's reply: The bimolecular rate equation expression for AOM applied in this study is the most frequently used AOM parameterization in reactive transport models (Regnier et al., 2011; *Earth-Science Reviews*). We followed the bimolecular approach because of the high uncertainty in half-saturation constants, in particular for the putative Fe-AOM pathway. For a detailed discussion about AOM parameterization, the referee is kindly referred to the study by Regnier et al. (2011; *Earth-Science Reviews*). We now justify our choice for the AOM parameterization in the revised manuscript **(lines 265-268)**.

*Table 6 + Fig 6: The labelling of the two kinds of methanogenesis is misleading. The light isotopic composition of methane implies that it is formed mainly through CO2 reduction rather than acetoclastic methanogenesis, i.e. that "Methanogenesis (OM)" is mainly CO2-based. "Methanogenesis (DIC)" is really a peculiarity of the model and completely and uniquely linked*

*to pyrite formation, so "Methanogenesis (FeS2)" would be more appropriate (but see also comment to Table 3 above).*

Author's reply: We agree with referee that reaction R6 in Table 3 and the labeling in Table 6 and Fig. 6 are difficult to understand and have improved it in the revised version (see the caption of Table 3). Furthermore, the two $CH_4$ pathways are now summed together in Table 6 and Fig. 6.

*Referee's comment: Fig. 7: Consider a colour version here. The darkest shading on the scale bars always appears darker than the darkest part of the figures. Because the shading varies so little from min. to max. it is very difficult to extract quantitative information.*

Author's reply: We have changed Fig. 7 to a color version.

**Referee # 3 (W.-L. Hong)**

*Referee's comment: Overall comments In the paper, Egger et al. present very comprehensive porewater and sediment geochemical data to discuss the cycles of C, S, Fe, and P in sediments that are unique in the sense of their depositional sequence and history. The authors collectively discuss the data with a rather complicated model, which is understandable due to the complexity of the system they work with. The paper is well-written and structured. The authors' attempt to elucidate the mass balance of several elements in this environment provide valuable insights to the coupling of these elements and the complexity of it. In general, the conclusions are convincing and well support by their data and modeling. As also a user of such transport-reaction models (Hong et al., 2014a&b; Torres et al., 2015; Hong et al., 2016), my major concern of the paper will be on the assumptions and setup of the model as well as several interpretations the authors made based on the model results. In conclusion, I think this is a very nice piece of work considering all the data and modeling work.*

Author's reply: We thank the referee for the compliment. We reply to the comments below and have revised the manuscript accordingly.

*Referee's comment: Major comments Modeling numerical issue: In conventional models for transportreaction models, advection (i.e. sediment burial) often inevitably results in numerical dispersion, concentration will decrease as time progresses with burial even without any reaction. This effect will be especially obvious when using high advection rate (burial rate), large time discretization, and a long modeling time. I've done some tests before (Hong et al., 2016 accepted by limnology and oceanography) by simulating time progression of a profile with sharp concentration change. After 140 years of simulation, the concentration is 20% reduced compared to the value it should have (see the attached file for this comparison). As for the sharp increase of OM content in your environment, you will inevitably encounter this numerical issue. I urge the authors to run some simulations with only burial (no diffusion and other reactions), and see how your sediment and porewater profiles will progress.*

Author's reply: We thank the referee for sharing his experience with us. The ReacTran package applied in this study (see line 285) accounts for numerical dispersion using total variation diminishing (TVD) slope limiters. A description of the ReacTran package can be found at: https://cran.r-project.org/web/packages/ReacTran/ReacTran.pdf

*Referee's comment: I also wonder what is the consequence to accelerate your model. The price of numerical issues will be greater when you accelerate it by using larger time steps and/or faster rates. It is almost no way to have a model that is both efficient and accurate. There is always a sacrifice.*

Author's reply: We could not find any indication that the model acceleration impacts the conclusions presented in this study. A higher sedimentation rate in the lake phase, however, largely improves the model efficiency as it significantly reduces the computing time.

*Referee's comment: The other potential numerical issue I want to point out for the authors is the convergence of the model results. You have to make sure you use temporal and spatial steps small enough so that the results are stable. This can be done by running the model several times (the same reactions and setup) with smaller time/space discretization for each run. Chose the smallest discretization that your model results stop changing.*

Author's reply: We used this approach in this study (i.e. we ran the model using various time and space discretizations). The model presented here thus reflects the smallest discretization for which the model results stop changing.

*Referee's comment: Conflict between observations and model results: In a few places in the paper, the authors didn't explain clearly the conflict between observations and model results. One example is the choice of chloride changes with time. The authors used a very different evolution pattern from what literature suggested because it provides a better fit of their chloride concentration. However, the time scale adopted by the authors (100yrs) is an order smaller than what is suggested in the literature (2000yrs). The authors provide no explanation about such difference. I envision that if the author use constant chloride from 2000yrs BP until now and increase fluid advection rate (larger u), they might be able to fit the profile. I think the authors should explain better why choosing such condition.*

Author's reply: We extended the total length of the model domain to 3000 cm (see lines 280-282), which allowed us to modify our salinization scenario to make it much more similar to that of Soulet et al. (2010) **(lines 293-298)**.

*Referee's comment: The other example is from line 369 to 371. The authors claimed the SR rates they estimated from the model in zone I & II are more correct the estimation from porewater profiles. This statement raises the question that, then how do you know the SR rate you estimated from these two zones are accurate since you have no data to support you.*

Author's reply: Sulfate reduction rates derived from pore water profiles represent net sulfate consumption, while rates derived from radiotracer injection and diagenetic modeling reveal total sulfate turnover in the sediment, i.e. gross sulfate reduction rates (e.g. Jørgensen (1978; *Geomicrobiol. J.*) and Jørgensen et al. (2001; *Deep Sea Res. Part I*)). Thus, modeled rates are generally higher than rates estimated from pore water profiles. We have clarified this in the revised manuscript **(lines 402-404)**. The referee is further kindly referred to lines 394-395 where we provide references to studies that have measured sulfate reduction rates in Black Sea sediments, which compare well with our model results.

*Referee's comment: Very high methanogenesis rate in sulfate reduction zone: In fig 6, there are two peaks of methane production (one in bottom water or first cm of sediment? While the other in zone II). My questions are two: 1) It is apparent that this methanogenesis is from OM decomposition. However, methanegenesis should be suppressed when the sulfate content is high, as in the case of zone II. I understand that although methanogenesis is inhibited by sulfate*

*content (E5 & E6 in Table 4), model can still produce very high ME rate when there is ultrahigh OM content. However, a model is a model, do you have any prove such high methanegenesis from your zone I and II. Considering the CH4 production rate and SO4-AOM rate from Fig.6. you should see either high methane or light d13C of methane in zone I and II if the rate is this high. I however don't quite see those from your profiles.*

Author's reply: The $CH_4$ production rate depicted in the original Fig. 6 is not correct, as it should be multiplied with the solid volume fraction of the sediment (increases from ~ 0.05 at the sediment surface to ~ 0.39 at depth) and divided by a factor of two (only 0.5 mole of $CH_4$ are produced per mole of organic matter during methanogenesis; see reaction R5 in Table 3; also see response to referee #2). Actual modeled rates of $CH_4$ production are thus < 30 pmol $cm^{-3}$ $d^{-1}$ in the marine deposits, which is an order of magnitude lower than modeled $SO_4$-AOM rates. Our modeled $CH_4$ production rates in the surface sediments (~ 7 pmol $CH_4$ $cm^{-3}$ $d^{-1}$) are still significantly lower than net rates of methanogenesis measured in surface sediments of the Peruvian margin of up to ~ 1 nmol $cm^{-3}$ $d^{-1}$ in the sulfate reduction zone, for example (Maltby et al., 2016; *Biogeosciences*).

*Referee's comment: 2) Back to fig6, your rates do not seem to balance. The highest CH4 production rate approaches 300 pmol/cm3/d which only stimulates an AOM rate less than maybe 20 pmol/cm3/d. If there is more production than consumption, isn't that you will methane accumulates in the porewater (i.e., high methane from that depth in the sediments). SR rate is over 2000 pmol/cm3/d in this section but sulfide production is only 300 pmol/cm3/d. where is the rest of sulfide production?*

Author's reply: We thank the referee for pointing out these inconsistencies. There has been an error in the plotting of the rates (see also answer to referee #2). The rate plots of Fig. 6 were not normalized to the same volume, meaning that the $SO_4$ reduction rate, $CH_4$ production and $S_0$ disproportionation should be multiplied by the solid volume fraction of the sediment, while rates of $SO_4$-AOM and Fe-AOM should be multiplied by the sediment porosity. In addition, only 0.5 mole of $CH_4$ are produced per mole of organic matter during methanogenesis (see reaction R5 in Table 3). We have corrected these errors in the revised manuscript (see Table 6 and Fig. 6).

*Referee's comment: The very complicated model: The authors use a rather complicated model in this study by choosing many reactions that are not totally necessary. For example, the authors choose to include aerobic processes (R1, R7-R12) and nitrate reduction (R2) even though there is no constraints on O2 and nitrate content in the porewater. I would also doubt the importance of these reactions due to the anoxic bottom water in Black Sea. The authors chose not to include Mn reduction due to its low content, which is fine with me, but decide to include all other processes that cannot be constrained? That is an odd decision to me. By excluding these unnecessary reactions, the authors can also improve the efficiency of the model.*

Author's reply: Oxic mineralization and nitrate reduction were implemented because of the oxic Black Sea lake phase. Bottom water concentrations of oxygen and nitrate for the Lake phase were taken from Reed et al. (2011; *GCA*). We agree with the referee that nitrate reduction plays only a minor role (also during the oxic Lake phase), but prefer to keep the reaction in the model, as removing it does not significantly improve the model efficiency.

*Referee's comment: I also wonder, with all the reactions assigned in the model, do the authors have enough constraints? I believe the answer should be close to yes as the authors have many data to support the model (which is very nice). I would urge the authors to spare a section in the text discussing the constraints for the model. To me, this is an extremely important but often ignored aspect in papers like this. I have done some initial analyses based on the reaction network in Table 3. For example, for Fe2+, the authors have R3, R9, R10, R13, R14, and R20 for sources, and R8, R15, R23, and R24 for sinks. Some source and sink terms may be constrained by the data of iron mineral speciation. When the same analyses being applied on HS in porewater, it seems like the abundance of different Fe-S minerals also depend on the source and sink terms of HS. A table such as tab6 but with more species included may be useful for such discussion.*

Author's reply: We kindly refer the referee to lines 239-241 and Table 5 where we describe the parameter constraints. We have now added a table with the mass balance for each species in the Supplementary Information **(see Supplementary Table S2)**.

*Referee's comment: One last comment on the complicated model, how does the model describe pH, which should be very important determining the type of dissolved sulfide and DIC. I don't see reactions such as H2S becomes HS-+H+ in Tab3 which describe the buffer capability of HS species (need same reactions for carbonate systems). Although there is usually no good constraint on pH, it's good to make sure pH falls in the right range especially when including pH-sensitive reactions.*

Author's reply: The model does not include pH, because it does not capture the precise underlying reaction mechanisms. Furthermore, we do not have pH data to compare the model results to. Adding pH would be a separate study in itself.

*Referee's comment: Minor comments Line 151: Please specify how you measure sulfide, phosphate, and DIC onboard.*

Author's reply: We have now added this information in the revised manuscript **(lines 157-162)**.

*Referee's comment: Line 211: why 20 meters? You should mark you explain this in the supplemental material.*

Author's reply: The referee is kindly referred to lines 279-280 and Supplementary Fig. S6. Note that it has changed to 30 m in the revised manuscript (see above).

*Line 255: is zero gradient a good assumption for methane? How do you know there is no deeper source of methane?*

Author's reply: We have no information about a potential deep source of $CH_4$ at our study site. However, the good fit between the modeled and measured ammonium profile (Fig. 3) indicates that it is likely that most of the $CH_4$ is produced within the model domain. We therefore think that a zero gradient assumption is a reasonable assumption for $CH_4$.

*Referee's comment: Line 289 to 293: You have same ammonium but higher methane in site4 and 5. Of course more severe degassing during core recovery in site 4 can be one explanation, but maybe there is more methane input from site 5 from greater depth. This echoes back my previous comment: is zero gradient really a good assumption for methane?*

Author's reply: If the difference in the $CH_4$ profiles between site 4 and 5 would be due to more $CH_4$ input from greater depth at site 5, it would imply that the measured concentrations at site 4 represent actual in-situ concentrations of pore water $CH_4$. However, we were not able to reproduce the observed ammonium profiles with such low rates of methanogenesis at depth. In contrast, our model suggests that in order to have sufficient pore water ammonium, $CH_4$ concentrations should be significantly higher than the measured concentrations at both sites. The high $CH_4$ concentrations derived from the model are also consistent with our observations of massive $CH_4$ degassing during coring and with previous observations in the western Black Sea shelf (Jørgensen et al., 2001; *Deep Sea Res. Part I*). We conclude that the ammonium profiles indicate that most of the $CH_4$ is produced within the model domain, rather than from greater depths, thus supporting our zero gradient assumption.

*Referee's comment: Line 304: How do you know the isotopic signature of methane is not affected by degassing?*

Author's reply: We base this conclusion on the smooth pore water profiles of $\delta^{13}C$-$CH_4$ and $\delta D$-$CH_4$ (see Fig. 4). To date, little is known about potential isotopic fractionation during degassing. However, it is thought that fast degassing is unlikely to result in major isotopic fractionation, as all $CH_4$ isotopes are lost simultaneously. We now clarify this in the revised manuscript **(line 328)**.

*Referee's comment: Line 533: Isn't that this will be capture in your orgP analyses?*

Author's reply: Mineral formation in microbial cells is not included in the model. Inclusions of Fe-P minerals initially formed in bacteria could be dissolved in the CDB step of the SEDEX extraction (Dijkstra et al., 2014, PLoS ONE).

*Referee's comment: Line 827: Do you have any constrain on C/P ratio? Maybe this explains why the fitting on porewater phosphate profile is not as good?*

Author's reply: The C/P ratio observed in the sedimentary record does not necessarily directly relate to the initial C/P ratio of the organic matter deposited on the seafloor due to the preferential regeneration of phosphorus from organic matter during anaerobic degradation (e.g. Ingall et al., 1993; *GCA*). It is thus of limited use as a constraint in this study.

*Referee's comment: Salinity/chloride: In many places of the paper, the authors mixed the term salinity and chloride concentration (e.g., line268-282). Of course these two properties are usually linear dependent on each other but they are fundamentally different and may correlate with each other very differently when Black Sea was more of a "lake" or a "Sea". I suggest the authors to use chloride concentration throughout the paper or explain how they convert salinity to chloride concentration.*

Author's reply: We have followed the referee's suggestion and explain how salinity is converted to chloride concentrations in the revised manuscript **(lines 297-298 and Supplementary Table S1)**.

*Referee's comment: FigS3: What is going on with the very high alkalinity at very top? where dic concentration looks normal...*

Author's reply: We thank the reviewer for pointing this out. After carefully checking the data, we have decided to omit them from the Supplementary Information due to uncertainty over data quality.

**List of all relevant changes made in the manuscript**

- Model sensitivity analysis to show the uncertainties associated with key processes (presented in the Supplementary Information, Figs S2 and S3)
- Model grid changed from constant to exponentially decreasing in order to speed up the sensitivity analyses (no effect found for model results)
- Revision of the salinization scenario to make it consistent with existing literature
- Providing information on how salinity was converted to $Cl^-$ and $SO_4^{2-}$
- Clarification of model formulation and corresponding results
- Mass balance for each species in the Supplementary Information (Table S2)
- More detailed explanation why we assume Fe-AOM as the only Fe reduction pathway at depth in the model
- Clarification of pore water subsampling methods
- More detailed information about methane saturation concentration and degassing of methane (also indicated in Figs 3 and 4)
- Correction of the rates shown in Fig. 6 and corresponding revision of Table 6
- Discussion of the limitation of the Rayleigh function to estimate fractionation factors in our study
- Showing the potential release of dissolved $Fe^{2+}$ via cryptic S cycling in our model
- Fig. 7 has changed to a color version
- More detailed discussion of sulfate and methane pore water profiles and corresponding fluxes of methane and sulfate to the SMTZ
- Reactivation of less reactive iron oxides highlighted as alternative iron reduction pathway

[revised manuscript text omitted]

---

## Author Response (AR2)

**Response to reviewers for revised manuscript bg-2016-64**

**Referee # 1 (Orit Sivan)**

*Referee's comment: Suggest to accept as is.*

Author's reply: We thank the referee for her insightful review that improved the quality of this paper.

**Referee # 2 (anonymous)**

*Referee's comment: The authors have done an excellent job in revising the paper in response to the reviewers' comments. Most importantly in my view, they have justified the assumptions and reasons for excluding organoclastic iron reduction in the methane zone, making it clear that the purpose of the study is to explore the potential for iron-dependent AOM rather than determining actual Fe-AOM rates. I only find one minor issue:*

*I still respectfully disagree with the use of the closed-system Rayleigh distillation model to determine fractionation factors for sulfate-AOM (l. 428-441). A remark is now included that determination of accurate factors require modelling, which is fine, but I am confused by the remark that "diffusion could be slower than oxidation". This true in the sense that some methanogenesis occurs in the AOM zone, but AOM rates in the SMTZ are about an order of magnitude higher than methane production rates there (Fig. 6), so most of the methane must be supplied by diffusion – this is by no means a "quasi-closed system" as claimed in the text.*

*Just a remark: I my comments to the 1st version I had missed the fact that hydrogenotrophic methanogenesis in the model depends on the DIC concentration and not on the hydrogen concentration. This is a strange formulation since methanogens in sediments usually deplete hydrogen to the energetic threshold (see papers by Hoehler et al.) and thus, expectedly, are hydrogen limited (with hydrogen supply limited by the degradation of organic matter and, possibly, pyrite formation). Thus, the modelling of two separate methanogenic pathways is questionable, but this has no real influence on the paper now, because the authors have chosen to present the combined methane production.*

Author's reply: We thank the referee for the compliment and for the critical remark regarding the modeling of hydrogenotrophic methanogenesis. We acknowledge the referee's view and have removed our previous interpretation of a "quasi-closed system" from the manuscript (lns 431-436).

**Referee # 3 (W.-L. Hong)**

*Referee's comment: The authors have addressed all points I raised previously. It's now almost flawless. I would however like to make a final note on the numerical issue (my very first point in the review). I believe ReacTran is doing a good job numerically but numerical error is simply inevitable in many cases. It's something all modelers should acknowledge and be honest about. A better understanding about how large such error might be is therefore very important. TVD is such a widely used scheme for solving transport-reaction equations and is also a good numerical scheme. It is however not immune from numerical issues if a proper discretization is not chosen. I encourage the authors to refer to studies such as Wang and Hutter (2001) (DOI: 10.1002/fld.197) or Toth and Odstrcil (1996) (doi:10.1006/jcph.1996.0197) which comparing different numerical schemes including TVD. As the authors replied: "The ReacTran package accounts for numerical dispersion using total variation diminishing (TVD) slope limiters". This statement does not guarantee everything. The only way to check numerical issues is to run the model with some clean scenario (such as advection of a square function).*

Author's reply: We thank the referee for the positive remarks and his helpful input regarding numerical errors. We will carefully consider the studies mentioned by the referee in our future modeling work.

[revised manuscript text omitted]